# A first-in-class inhibitor of Hsp110 molecular chaperones of pathogenic fungi

Liqing Hu [1,6,7,8], Cancan Sun[1,8], Justin M. Kidd[1], Jizhong Han[2], Xianjun Fang[3], Hongtao Li[1], Qingdai Liu[4], Aaron E. May[5], Qianbin Li[6], Lei Zhou [2] ✉ & Qinglian Liu [1] ✉

Proteins of the Hsp110 family are molecular chaperones that play important roles in protein homeostasis in eukaryotes. The pathogenic fungus *Candida albicans*, which causes infections in humans, has a single Hsp110, termed Msi3. Here, we provide proof-of-principle evidence supporting fungal Hsp110s as targets for the development of new antifungal drugs. We identify a pyrazolo[3,4-*b*] pyridine derivative, termed HLQ2H (or 2H), that inhibits the biochemical and chaperone activities of Msi3, as well as the growth and viability of *C. albicans*. Moreover, the fungicidal activity of 2H correlates with its inhibition of in vivo protein folding. We propose 2H and related compounds as promising leads for development of new antifungals and as pharmacological tools for the study of the molecular mechanisms and functions of Hsp110s.

*Candida albicans* is an opportunistic pathogen that can cause various fungal infections (candidiasis), ranging from oral thrush to candidemia, one of the most prevailing disseminated bloodstream infections (with a mortality rate of over 40%)[1,2]. Although *C. albicans* is typically commensal for healthy people[3,4], it is one of the leading causes of hospital-acquired infections and can cause serious infections for critically ill or immunocompromised individuals, such as patients with AIDS, cancers, organ transplants, or implants[5-7]. However, only three classes of antifungal drugs targeting two unique pathways (sterol ergosterol and the cell wall) are available to treat candidiasis[8-10]. Moreover, the treatment has been further complicated by a dramatic rise in resistance to these available antifungal drugs[8-10]. Thus, efficient therapeutic options with novel modes of action are needed urgently for treating this dangerous pathogen.

110 kDa Heat Shock Proteins (Hsp110s) are found in the cytosol of eukaryotes and play essential roles in maintaining cellular protein homeostasis (proteostasis)[11-20]. Msi3 is the only Hsp110 in *C. albicans*,

and it is essential for the survival, growth, and infection of *C. albicans* in human hosts[19,21,22]. In contrast, Hsp110s are not essential for mammals under normal conditions, as suggested by knockout experiments using mice[23-25]. Moreover, the sequence conservation between human and fungal Hsp110s is relatively low (<40% identity). As a result, Msi3 may represent a favorable target for designing novel and effective therapeutics with fewer side effects for candidiasis. However, no inhibitors have been reported for any fungal Hsp110s. In fact, it has been challenging to develop modulators for Hsp110s in general due to a limited understanding of the molecular mechanisms of their chaperone activity, although inhibiting Hsp110s has been established as a potentially effective strategy against cancers[26-28]. To date, only one specific small molecule inhibitor has been reported for Hsp110s[29], and this inhibitor is a competitor of ATP for a human Hsp110, which raises questions regarding specificity.

Hsp110s are a unique class of molecular chaperones. They act as both independent chaperones and cochaperones for Hsp70s, another

[1]Department of Physiology and Biophysics, School of Medicine, Virginia Commonwealth University, Richmond, VA 23298, USA. [2]Institute of Molecular Physiology, Shenzhen Bay Laboratory, Shenzhen 518107, Guangdong, China. [3]Department of Biochemistry and Molecular Biology, School of Medicine, Virginia Commonwealth University, Richmond, VA 23298, USA. [4]Key Laboratory of Food Nutrition and Safety, Tianjin University of Science & Technology, Tianjin 300457, China. [5]Department of Medicinal Chemistry, School of Pharmacy, Virginia Commonwealth University, Richmond, VA 23298, USA. [6]Present address: Department of Medicinal Chemistry, Xiangya School of Pharmaceutical Sciences, Central South University, Changsha 410013 Hunan, China. [7]Present address: Key Laboratory of Study and Discovery of Small Targeted Molecules of Hunan Province, Department of Pharmacy, School of Medicine, Hunan Normal University, Changsha, Hunan, China. [8]These authors contributed equally: Liqing Hu, Cancan Sun. ✉e-mail: zhoulei@szbl.ac.cn; qinglian.liu@vcuhealth.org

essential class of molecular chaperones[30–32]. In fact, Hsp110s are distant homologs of Hsp70s. By themselves, Hsp110s have a uniquely prominent chaperone activity in preventing aggregation of denatured proteins (i.e., "holdase activity")[33–37]. However, they do not share the hallmark activity of Hsp70s in directly assisting protein folding. As cochaperones, Hsp110s function as the major nucleotide-exchange factors (NEFs) for cytosolic Hsp70s, facilitating the exchange of ADP for ATP in Hsp70s[13,14,38–40]. Various studies have suggested that Hsp110s participate in almost all known processes associated with cytosolic Hsp70s, including de novo protein folding and refolding under stress, protein transportation into the endoplasmic reticulum, solubilization of protein aggregates, and protein degradation[11,15,17–19,34,35,41–53]. While the function and mechanism of the NEF activity are well established, the functions and biological roles of the holdase activity remain enigmatic.

The unique chaperone activity of Hsp110s is rooted in their distinctive biochemical and structural properties. Hsp110s contain two functional domains: a nucleotide-binding domain (NBD) and a substrate binding domain (SBD). The NBD binds to ATP, which is crucial for both the NEF and holdase activities[16,54–57]. While it has been postulated that the SBD binds to polypeptide substrates, both the substrate binding site and properties remain largely a mystery. Our current biochemical and structural understanding of Hsp110s is primarily based on studies of Sse1, an Hsp110 from *Saccharomyces cerevisiae*[13,14,16,19,34,36,49,54,55,58–64]. Previously, we have solved the first X-ray crystal structure of a full-length Hsp110 using Sse1, revealing extensive NBD-SBD contacts in the ATP-bound state[54]. In addition, two structural studies on Sse1 in complex with Hsp70s have provided insight into the molecular mechanism of its NEF activity[60,61]. To understand the unique chaperone activity of Hsp110s, we recently purified Msi3 and characterized its biochemical properties[65,66].

In this work, we have taken advantage of the unique biochemical properties of Msi3 and screened an innovative chemical library published recently[67]. Excitingly, we have identified a novel inhibitor of Msi3, designated HLQ2H (shortened as 2H), which strongly inhibits both the biochemical and chaperone activities of Msi3. Interestingly, the holdase activity is almost completely abolished by 2H while the NEF activity is left largely intact. These results provide the first direct evidence to support the importance of the elusive holdase activity of Hsp110s in protein folding. Importantly, 2H shows a strong inhibition on the growth and viability of *C. albicans* while having limited toxicity on three human cell lines. Therefore, 2H is an invaluable tool for exploring the elusive mechanisms and functions of Hsp110s and for developing novel antifungals.

## Results
### Identification of 2H, a novel inhibitor for Msi3, the sole and essential Hsp110 in *Candida albicans*
One of the major challenges for inhibitor identification is due in part to a lack of a robust, reliable biochemical assay for screening inhibitors for Hsp110s directly. To search for specific inhibitors for Msi3, we took advantage of its unique biochemical properties. We recently discovered that Msi3 has a high affinity for binding ATP, and this high-affinity binding is controlled by the allosteric coupling between its two functional domains[65,66]. Compounds that inhibit the high-affinity binding of ATP to Msi3 could be caused by either directly disrupting ATP binding to the nucleotide-binding site or influencing allosteric coupling. Thus, this high-affinity ATP binding may allow us to discover modulators with three major modes of inhibition: 1) direct competition with ATP binding, 2) direct modulation of allosteric coupling, and 3) indirect influence of allosteric coupling by affecting the SBD activity. The latter two modes are particularly interesting because of their potential specificity to Msi3. To monitor the high-affinity binding of ATP to Msi3, we set up a fluorescence polarization assay using ATP-FAM, a fluorescently labeled ATP. This robust and reproducible assay

(Z-factor = 0.82) enabled us to screen a recently published compound library including 23 novel compounds designed based on the chemical structures of Riociguat and Compound C[67]. Fortunately, we found four structural-related compounds that reduced the ATP binding to Msi3 by more than 50% (Fig. 1a). Among them, 2H showed the most reduction in ATP binding and was therefore selected as the focus of the further investigation. Figure 1b shows the chemical structure of 2H.

To confirm the inhibition of ATP binding by 2H, we determined the dissociation constants ($K_d$) of Msi3 in the presence and absence of 2H. As shown in Fig. 1c, treatment with 2H at 100 μM reduced the affinity of Msi3 for ATP by about 6-fold. In addition, 2H showed a similar inhibition on Sse1 (Fig. S1a), the major Hsp110 from *S. cerevisiae*, which is consistent with the high degree of sequence and function conservation between Msi3 and Sse1[66]. To assess the specificity of 2H, we purified and tested Hsp105, the primary human Hsp110, and Ssa1, a fungal Hsp70. 2H had little effect on ATP-FAM binding to Hsp105 (Fig. S1b), consistent with the low degree of sequence conservation among Hsp110s (Fig. S2). Moreover, Ssa1 is the major cytosolic Hsp70 in yeast and is highly conserved (84.9% sequence identity between *S. cerevisiae* and *C. albicans*). As shown in Figs. S1c, 2H did not significantly impact ATP-FAM binding to Ssa1 from *S. cerevisiae*, which is in line with the sequence and function differences between Hsp70s and Hsp110s although they are homologs. In addition, since 2H was designed and synthesized based on the chemical structures of Riociguat and Compound C[67], we used these two chemicals as control compounds. We also tested fluconazole, one of the most widely used antifungal agents[68,69], as an additional control. As shown in Fig. S3a, these compounds had limited influence at the same concentration. Taken together, our results suggest that 2H may represent a novel class of inhibitors specific to fungal Hsp110s.

### 2H reduces the ATP binding affinity of Msi3 through a novel mode
To determine if 2H inhibits ATP binding to Msi3 by competing directly with ATP for the nucleotide-binding site, we took advantage of the fluorescence spectrum of 2H. As shown in Figs. 1d, 2H exhibited a strong fluorescence spectrum upon excitation at 308 nm ($\lambda_{em} = 432.3 \pm 1.8$ nm). Interestingly, adding Msi3 not only increased the fluorescence intensity of the 2H spectrum by a factor of $1.63 \pm 0.20$, but also shifted the spectrum to a shorter wavelength (Fig. 1d and S4a, $\lambda_{em} = 420.5 \pm 3.4$ nm), suggesting a direct binding of 2H to Msi3. As a protein control, lysozyme was incubated with 2H, and the fluorescence spectrum did not show an appreciable difference from that of 2H alone (Fig. S4b). Moreover, incubation of Msi3 with ATP prior to 2H treatment further increased the fluorescence intensity of 2H (by a factor of $2.84 \pm 0.40$, $\lambda_{em} = 423.1 \pm 1.9$ nm), whereas ATP alone showed little alteration to the spectrum of 2H ($\lambda_{em} = 431.8 \pm 1.9$ nm). Since the ATP concentration used (2 mM) was much higher than that of 2H (5 μM), 2H must bind Msi3 in the presence of ATP. Thus, 2H most likely inhibits the ATP binding to Msi3 not through direct binding to the nucleotide-binding site. Instead, 2H may reduce the ATP binding affinity by either regulating allosteric coupling or affecting the activity of the SBD. The enhanced fluorescence intensity of 2H after Msi3 was incubated with ATP suggests that ATP may influence the binding of 2H to Msi3.

To investigate whether 2H directly influences the allosteric coupling in Msi3, we carried out limited trypsin digestion. Like Sse1, Msi3 has two overall conformations: the ATP-bound and ADP-bound/ nucleotide-free (apo) states[65,66]. As shown before, Msi3 is more sensitive to trypsin in the absence of nucleotide (apo state) than in the presence of ATP (Fig. 2a). This is likely due to the extensive NBD-SBD contacts, as observed in the crystal structures of the ATP-bound Sse1[54,60,61,65,66]. While it has been suggested that the ADP-bound and nucleotide-free states have different conformations, no structures have been solved for either state[14,16,19,36,54,55,57,63,70]. Our analysis showed that 2H treatment did not significantly affect the tryptic digest patterns

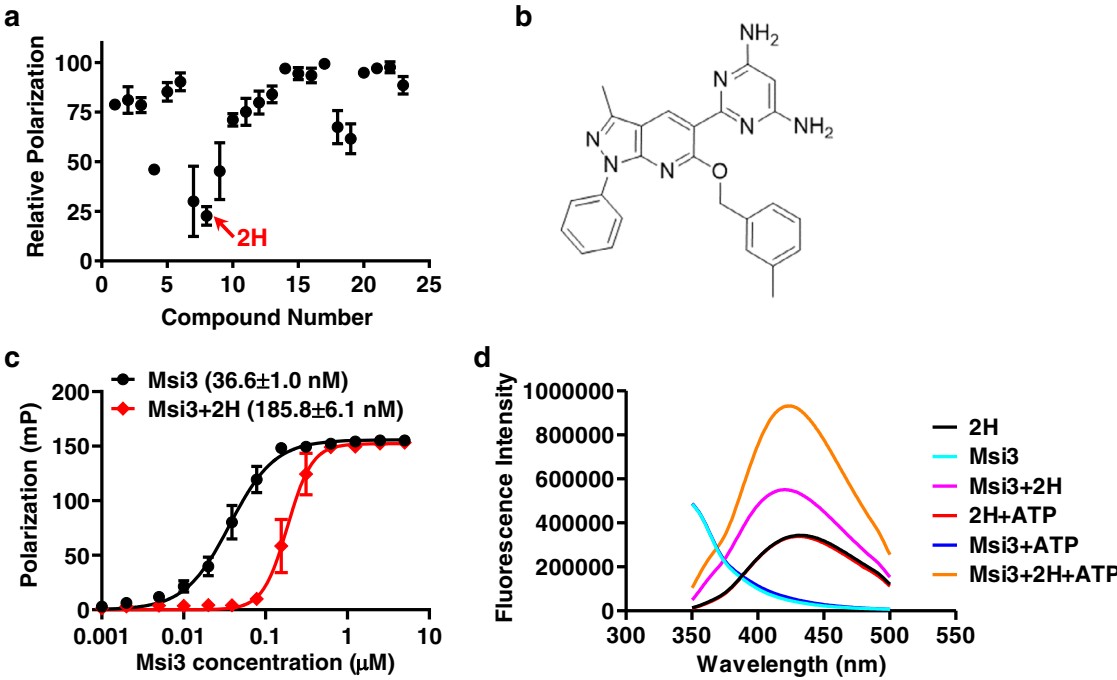

**Fig. 1 | 2H was identified as an inhibitor of purified Msi3 protein. a** The discovery of 2H through a chemical screen. A collection of chemicals was screened for their ability to inhibit ATP-FAM binding to Msi3 in a fluorescent polarization assay. The polarization reading is an indication of ATP-FAM binding to Msi3. Relative polarization was plotted after setting the readings of ATP-FAM and ATP-FAM + Msi3 as 0 and 100, respectively. 2H is highlighted by a red arrow. Data are presented as mean values + /- SEM (*n* = 5 independent experiments). **b** The chemical structure of 2H. **c** The inhibition of 2H on the ATP binding affinity of Msi3. Dissociation constants ($K_d$) are listed in parentheses. A fluorescent polarization assay was carried out resembling that in (**a**). Data are presented as mean values + /- SEM (*n* = 3 independent experiments). **d** The binding of 2H to Msi3 in the presence and absence of ATP. The fluorescence spectra of 2H were recorded in the presence and absence of Msi3 with the excitation wavelength set at 308 nm. For +ATP samples, ATP was included at a final concentration of 2 mM. Source data are provided as a Source Data file for **a**, **c**, and **d**.

of Msi3 (Fig. 2a), suggesting that 2H has minimal impact on the overall allosteric coupling of Msi3 upon ATP binding.

We next analyzed the effect of 2H on the activity of the SBD in binding peptide substrates. Previously, we have determined the peptide substrate binding affinity of Msi3 using the TRP2-181 peptide, a model peptide substrate that has been shown to bind to Hsp110s including Msi3 and Sse1[36,65]. As illustrated in the top panel of Fig. 2b, incubating Msi3 with 100 μM 2H reduced its affinity for the TRP2-181 peptide by nearly 9-fold. Moreover, consistent with the reduction in affinity, 2H efficiently competed off the TRP2-181 peptide that had initially been bound to Msi3 (as shown in the bottom panel of Fig. 2b). In contrast, 2H did not show substantial influence on the substrate binding activity of either Hsp105 or Ssa1 (Fig. S1d, e). In addition, neither Riociguat, Compound C, nor fluconazole exhibited significant effect on Msi3-TRP2-181 binding (Fig. S3b). These findings are consistent with the above results of our ATP-FAM binding assays. Taken together, the reduced ATP affinity caused by 2H could be partly due to the reduced affinity of substrate binding to the SBD.

Consistent with the limited trypsin digestion results, 2H showed limited impact on the NEF (nucleotide exchange factor) activity of Msi3 (Fig. 2c and S5). Hsp110s such as Msi3 and Sse1 are known to serve as the major NEFs for cytosolic Hsp70s[13–15,19,39,41,65,66]. Previously, using ATP-FAM in a fluorescence polarization assay, we have reported that Msi3 facilitates nucleotide exchange of Ssa1, the major Hsp70 in *S. cerevisiae*[65,66]. In this assay, an Ssa1-ATP complex was formed using ATP-FAM; then, the release of the pre-bound ATP-FAM was monitored over time. The addition of Msi3 alone had little influence on the release of ATP-FAM from Ssa1 (Fig. 2c and S5). The addition of regular ATP resulted in reduced polarization values over the time of incubation, suggesting that the pre-bound ATP-FAM was slowly replaced by the regular ATP. In contrast, adding Msi3 together with

regular ATP drastically sped up the reduction of polarization, consistent with the reported NEF activity for Msi3. 2H, both alone and in the presence of Msi3, showed limited influence on the nucleotide release of Ssa1 (Fig. 2c, less than 20% reduction in $k_{off}$ in the presence of ATP). Our native gel analysis on the Msi3-Ssa1 complex formation provided further support for the lack of significant impact on the NEF activity. The formation of the Msi3-Ssa1 complex is consistent with the NEF activity. As shown in Fig. 2d, Msi3 forms a stable complex with Ssa1 on a native gel regardless of the presence of 2H. It was shown that the ATP binding and ATP-bound state of Hsp110s is required for the NEF activity[13–16,19,39,41,54–57]. The insignificant impact of 2H on the NEF activity of Msi3 is consistent with the above results suggesting that 2H has little impact on the allosteric coupling of Msi3 (Fig. 2a) and does not compete directly with ATP for the nucleotide-binding site (Fig. 1d). In addition, since the peptide substrate binding is reduced significantly by 2H, it is likely that the NEF activity is independent of the peptide substrate binding activity of the SBD.

Taken together, 2H most likely inhibits the ATP binding affinity of Msi3 by affecting peptide substrate binding. This mode is different from the previously published inhibitor for human Hsp110[29], which competes directly with ATP for the nucleotide-binding site. Therefore, 2H interacts and inhibits Msi3 through a novel mode.

## 2H abolishes the in vitro chaperone activities of Msi3

As an Hsp110, Msi3 has two known in vitro chaperone activities: preventing protein aggregation (i.e., the holdase activity) and assisting Hsp70s in protein folding[13,14,33–37]. We tested whether 2H impacts either of these activities.

To determine the impact of 2H on the holdase activity of Msi3, we used a previously reported assay with purified firefly luciferase as a model substrate[33–37]. In this assay, upon incubation at 42 °C, luciferase

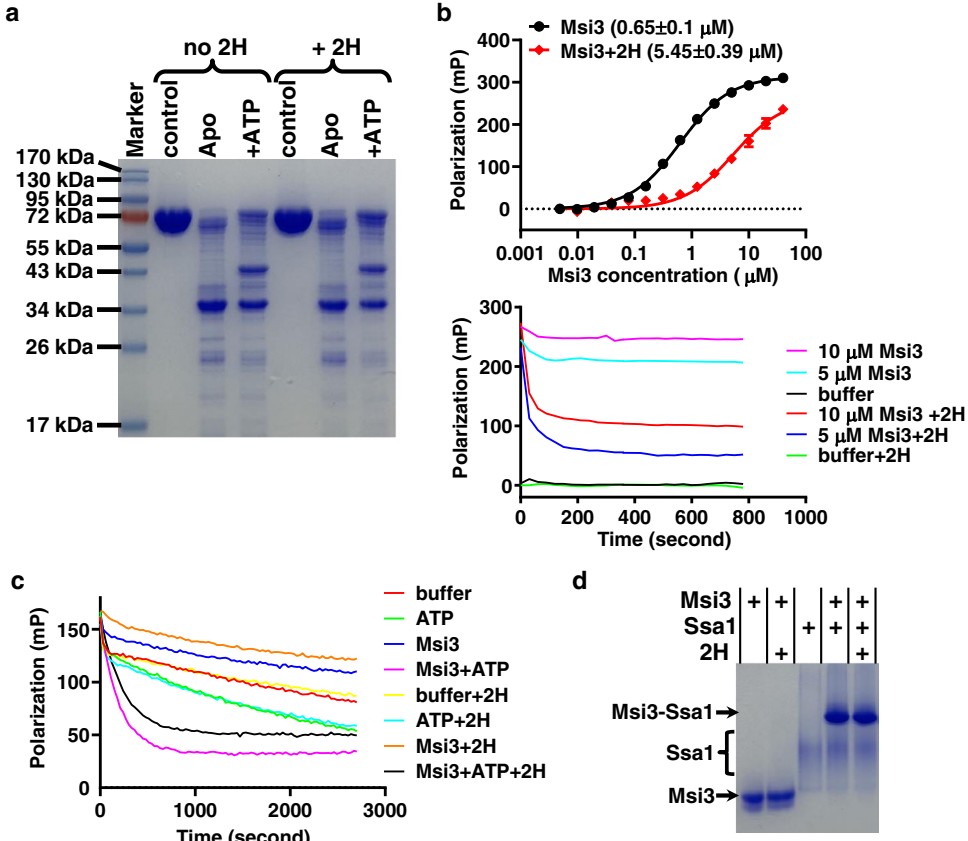

**Fig. 2 | 2H influenced the biochemical activities of Msi3 in different ways. a** The ATP-induced allosteric coupling assayed by limited trypsin digest. control: trypsin was not added. Apo: nucleotide was not added. +ATP: ATP was included at a final concentration of 2 mM. **b** The effect of 2H on the peptide substrate binding activity of Msi3. A fluorescent polarization assay was carried out using the fluorescently labeled TRP2-181 peptide as a substrate. Top: binding curves with dissociation constants ($K_d$) listed in parentheses. The polarization value is an indication of Msi3 binding to the TRP2-181 peptide. Data are presented as mean values +/- SEM ($n = 3$ independent experiments). Bottom: competition assay with 2H. Msi3 was first incubated with the TRP2-181 peptide to allow binding, and then 2H was added to compete off the pre-bound TRP2-181 peptide. Polarization values were monitored over time, with the concentrations of Msi3 labeled. Reactions without 2H were used as controls. buffer: the TRP2-181 peptide only. **c** NEF activity of Msi3 on Ssa1. After a complex formed between Ssa1 and ATP-FAM, the reduction of polarization values represented the release of ATP-FAM from Ssa1 upon the addition of ATP or/and Msi3. Release kinetics ($k_{off}$) are summarized in Fig. S5. **d** Formation of the Msi3-Ssa1 complex. Complex formation was analyzed using a native polyacrylamide gel electrophoresis (PAGE), then visualized after staining with Coomassie blue. For all assays, 2H was added at a concentration of 100 μM unless labeled otherwise. Source data are provided as a Source Data file for all the panels.

denatures and aggregates, resulting in an increase in OD readings at 320 nm ($OD_{320}$) over time (Fig. 3a). As a chaperone, Msi3 is resistant to heat treatment, indicated by a limited increase in $OD_{320}$ (i.e., aggregation). 2H showed little impact on the aggregation of luciferase or Msi3 alone. The addition of Msi3 drastically reduced the aggregation of luciferase, consistent with the holdase activity. Excitingly, 2H treatment almost completely abolished the holdase activity of Msi3 (Fig. 3a). To confirm the inhibition of 2H on the holdase activity of Msi3, we tested a previously characterized Msi3 mutant I164D[65]. Ile164 is located on the NBD-SBD interfaces. Our previous biochemical analyzes have shown that the I164D mutant disrupts the NBD-SBD allosteric coupling and abolishes the NEF activity while leaving the holdase activity largely intact. A similar inhibition of holdase activity was observed for this mutant when treated with 2H (Fig. S6a). Furthermore, we have identified Ulp1, a protein from *S. cerevisiae*, as another model substrate for Msi3's holdase activity, although higher concentrations of Ulp1 and Msi3 are required than those for the assay using luciferase (Fig. S6b). Importantly, a similar inhibition of the holdase activity was observed for 2H (Fig. S6b), supporting that the inhibition of the holdase activity of Msi3 by 2H is general rather than substrate specific.

To determine the $IC_{50}$ for 2H inhibition of Msi3's holdase activity, we employed a previously reported assay that couples the holdase activity of Hsp110 with the refolding activity of the Hsp70/Hsp40 chaperone system[29], thereby circumventing the need for large amounts of proteins required by the $OD_{320}$ method described above. In this alternative assay, luciferase was first denatured by incubating at 42 °C, which leads to its aggregation as described above. When incubated with an Hsp70·Hsp40 chaperone pair, the denatured luciferase failed to refold and regain activity due to aggregation (Fig. S7a). In this study, Ssa1 and Ydj1 were used. Ssa1 is the major Hsp70 in *S. cerevisiae*, and Ydj1 is an Hsp40 partner of Ssa1 for protein folding. In contrast, when Msi3 was added during the denaturation at 42 °C to prevent aggregation, the denatured luciferase regains activity upon incubating with Ssa1 and Ydj1 (Fig. S7a). Using this assay, the $IC_{50}$ for 2H was calculated to be $5.02 \pm 0.35$ μM (Fig. 3b). When analyzing the effect of compound controls (Fig. S7b), little to no impact was observed for either Compound C or fluconazole while the limited impact was seen for Riociguat (estimated $IC_{50} > 60$ μM). In addition, we analyzed the impact of 2H on human Hsp105. A much weaker inhibition was observed with an estimated $IC_{50}$ over 40 μM (Fig. 3b), consistent with the low sequence conservation between human Hsp105 and Msi3 (Fig. S2). Furthermore, while Msi3 can functionally substitute Sse1, human Hsp105 failed to do so despite significant expression (Fig. S8). Taken together, the inhibition of the holdase activity of Msi3 is specific to 2H.

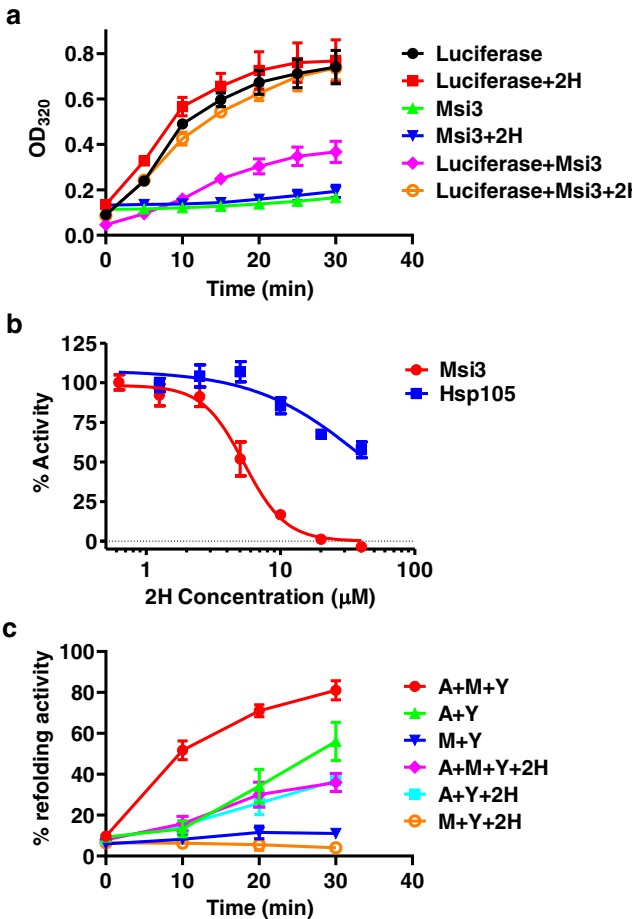

**Fig. 3 | 2H compromised the chaperone activities of purified Msi3 protein. a** The holdase activity of Msi3 assayed using OD readings at 320 nm (OD$_{320}$). Purified firefly luciferase was used as a substrate. The OD$_{320}$ values were recorded during incubation at 42 °C. Increased OD$_{320}$ values represent the aggregation of luciferase. Data are presented as mean values +/- SEM ($n = 3$ independent experiments). **b** The effect of 2H on the holdase activity of Msi3 (red circles) and Hsp105 (green squares). The substrate is firefly luciferase. The activity in the absence of 2H was set as 100%. Data are presented as mean values +/- SEM ($n = 7$ independent experiments). **c** Refolding activity of the Ssa1 chaperone machinery. Firefly luciferase was used as a substrate. After denaturation, the refolding of luciferase in the presence of different combinations of chaperones was measured over time. Chaperones included Ssa1 (A), Msi3 (M), and Ydj1 (Y). The activity of luciferase before denaturation was set as 100%. Data are presented as mean values +/- SEM ($n = 3$ independent experiments). Source data are provided as a Source Data file for all the panels.

In the next step, we investigated the effect of 2H on the activity of Msi3 in assisting the refolding activity of Hsp70 and Hsp40 (Fig. 3c). As previously reported[13,14], Ssa1 and Ydj1 together were able to refold heat-denatured luciferase modestly (Fig. 3c). However, the addition of Msi3 significantly improved the refolding activity[36,65,66]. Importantly, this Msi3-dependent refolding activity was effectively eliminated following 2H treatment, whereas the impact of 2H on the refolding activity of Ssa1/Ydj1 was limited (Fig. 3c). Two types of control tests were carried out to evaluate the specificity of 2H's inhibition. The first type was designed to test the effect of 2H on a human Hsp70-Hsp110-Hsp40 chaperone system. We have purified the corresponding human chaperones Hsp70, Hsp105, and HDJ2, which showed similar refolding activity as that of the Ssa1-Msi3-Ydj1 chaperone system. A limited impact was observed for 2H on this human chaperone system (Fig. S9a), consistent with the weak inhibition of 2H on the holdase activity of Hsp105. In the second type of control, we used three compounds (fluconazole, Riociguat, and

**Table 1 | The effect of 2H on the growth and viability of various yeast strains/isolates**

| Yeast strains/isolates | MIC$_{90}$−2H | MFC-2H | MIC$_{90}$-FLC |
|---|---|---|---|
| *C. albicans* SC5314 | 25 | 25 | 6.4 |
| *C. albicans* SC5314 + sorbitol | 25 | 25 | N/A |
| *C. albicans* JS14 | 25 | 25 | 500 |
| *C. albicans* JS15 | 25 | 25 | >500 |
| *C. albicans* FH5 | 12.5 | 25 | 125 |
| *C. albicans* 12-99 | 25 | 25 | >500 |
| *C. glabrata* 2001 | 25 | 25 | 500 |
| *S. cerevisiae* BY4742*MSI3 sse1Δ* | 15 | 20 | N/A |

The MIC$_{90}$ (MIC$_{90}$ −2H) and MFC (MFC-2H) of 2H were determined on several yeast strains/isolates. Fluconazole (MIC$_{90}$-FLC) was used as a control. MIC$_{90}$: the minimum inhibitory concentration for 90% inhibition. The minimal fungicidal concentration (MFC): the minimal concentration of the drug resulting in less than 0.1% of the cells alive relative to the original inoculum. *N/A* not determined.

Compound C) as controls. As shown in Fig. S9b-d, none of these compounds significantly affected the Ssa1-Msi3-Ydj1 chaperone system. Taken together, 2H specifically inhibits the Msi3-dependent refolding activity of the Hsp70 chaperone system.

The most remarkable feature of 2H is its ability to eliminate the holdase activity of Msi3 while leaving the NEF activity largely intact. This is the first instance where the holdase activity of an Hsp110 has been substantially reduced without affecting the NEF considerably. Therefore, 2H is a unique compound and could be an invaluable tool for probing the mechanism and in vivo function of Hsp110s, including Msi3.

### 2H effectively reduces the growth of *Candida albicans* and is fungicidal

Since Msi3 is essential for the growth and viability of *C. albicans*[19,21,22], we analyzed the effect of 2H on this fungal species. We first tested the widely used wild-type *C. albicans* strain SC5314. As shown in Table 1 and Fig. 4a, the growth of SC5314 was significantly inhibited by 2H with an MIC$_{90}$ (minimum inhibitory concentration for 90% inhibition) of 25 μM. Fluconazole (FLC), the most common over-the-counter antifungal[68,69], was used as a control, and the MIC$_{90}$ value obtained in our study was consistent with SC5314 being a fluconazole-sensitive strain (Table 1 and Fig. 4a). Neither Compound C nor Riociguat showed notable effect on the growth of SC5314 (Fig. S10), consistent with their lack of significant impact on either the biochemical or chaperone activity of Msi3. This is further supported by the fact that neither compound has been documented in published screens for antifungals.

Antifungals can be either fungistatic (inhibits growth) or fungicidal (kills the pathogen). Azoles, such as fluconazole, are one of four classes of approved antifungals[8–10]. However, azoles are normally fungistatic, and prolonged usage results in faster resistance development than that of fungicidal agents[9,71]. Importantly, 2H was shown to be fungicidal toward SC5314 with MFC (minimum fungicidal concentration, the minimum concentration for killing >99.9% cells) at 25 μM (Table 1). Furthermore, time-kill curves demonstrate that 2H exerts its fungicidal effect within one hour of treatment at concentrations of 50 and 100 μM, the 2- and 4-fold concentrations of the MFC, respectively, which are used by standard to determine time-kill curves (Fig. 4b). At the MFC concentration (25 μM), the rate of 2H-induced reduction in viability is markedly lower. Fluconazole was tested as a control. Consistent with published results, it showed little impact on viability. This suggests that resistance to 2H may develop at a slower rate.

Since azoles and 2H inhibit *C. albicans* through different mechanisms, we investigated whether there is synergistic effect

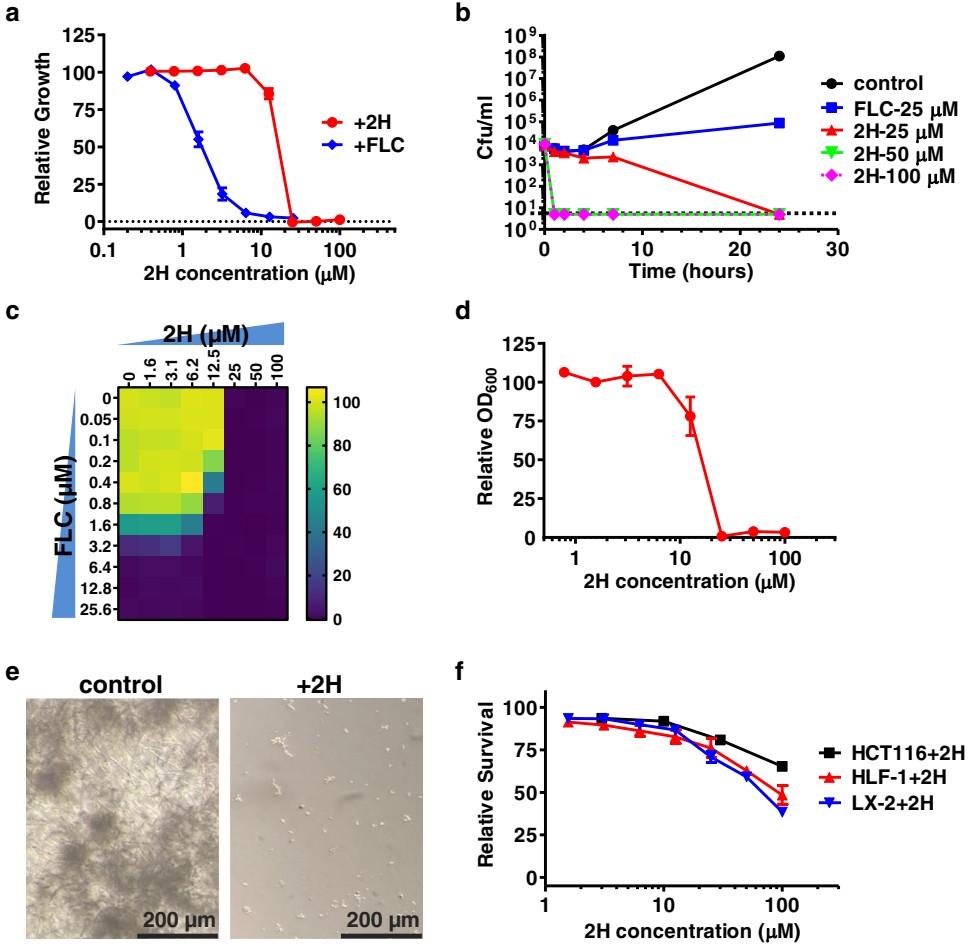

**Fig. 4 | 2H showed limited impacts on human cells while inhibiting the growth and biofilm formation of C. albicans. a** 2H inhibits the growth of C. albicans strain SC5314. Fluconazole (FLC) was used as a control. Relative growth was determined by setting the growth in the absence of compounds as 100%. Data are presented as mean values +/- SEM ($n$ = 8 for 2H and 9 for FLC independent experiments). **b** 2H time-kill curves of the *C. albicans* strain SC5314. The black dotted line indicates the detection limit (>99.9% reduction compared with the original inoculum). The growth of SC5314 with no treatment (control) or fluconazole (FLC) treatment were determined for comparison. Cfu: colony forming unit. Data are presented as mean values +/- SEM ($n$ = 3–6 independent experiments). **c** Checkerboard analysis of antifungal activity for a combination of 2H and fluconazole (FLC). Relative growth (calculated by setting the growth without compounds as 100%) is displayed in heat-map format. The plot is an average of 6 independent experiments ($n$ = 6). Color scale for relative growth is provided on the right. **d** The effect of 2H on biofilm formation in vitro. After biofilm formation, the plates were read at $OD_{600}$. The relative $OD_{600}$ values were calculated by setting the biofilm formation without any treatment as 100%. Data are presented as mean values +/- SEM ($n$ = 3 independent experiments). **e** Representative images of biofilm formation with (right) and without (left) 2H treatment. Images were captured under light microscopes. 2H was used at a concentration of 25 μM. **f** The effect of 2H on human cells. HCT116: a human colorectal cancer cell line; HLF-1: a human lung fibroblast cell line; LX-2: human hepatic stellate cell line Lieming Xu-2. Relative survival was determined by setting the survival in the absence of 2H as 100%. Data are presented as mean values +/- SEM ($n$ = 3 independent experiments). Source data are provided as a Source Data file for all the panels.

between 2H and fluconazole on the growth of SC5314 when administered together. We carried out a chequerboard (or checkerboard) assay and determined the fractional inhibitory concentration index (FICI) to be $0.91 \pm 0.03$ ($n$ = 6, range: 0.82–1.00) (Fig. 4c). Thus, no apparent synergistic effect was observed between 2H and fluconazole.

Next, we analyzed the effect of 2H on several fluconazole-resistant *C. albicans* isolates since resistance to fluconazole and other azoles has become increasingly problematic in treating infections caused by *C. albicans*. Four strains isolated from patients with either vaginal infections, AIDS, or marrow transplant were tested: JS14, JS15, FH5, and 12-99[72–74] (kindly provided by Dr. Theodore White). As shown in Table 1, similar $MIC_{90}$ and MFC values were observed across all these strains. Consistent with their reported resistance to fluconazole, the $MIC_{90}$ value of fluconazole for each of these strains was much higher than that of SC5314, a fluconazole-sensitive strain. Furthermore, a chequerboard analysis on JS14 suggested no clear synergy between 2H and fluconazole (FICI = $1.01 \pm 0.06$, $n$ = 6, range: 0.91–1.08).

In addition, we tested the effect of 2H on biofilm formation in vitro to estimate its potential as an antifungal. As shown in Fig. 4d, biofilm formation is effectively suppressed by 2H at concentrations higher than 25 μM. Examination of the plates under light microscope indicated that hyphae formation was visibly impaired by 2H treatment (Fig. 4e).

## 2H has limited impact on human cells but a fungicidal effect on *Candida glabrata*

To evaluate the impact of 2H on human cells, we analyzed its effect on three different human cell lines: HCT116, HLF-1, and LX-2. HCT116, a human colorectal cancer cell line, was used as a reference as a previously published human Hsp105 inhibitor showed significant inhibition on this cell line[29]. HLF-1 and LX-2 are a human lung fibroblast cell line and hepatic stellate cell line Lieming Xu-2, respectively. As shown in Fig. 4f, we observed much weaker growth inhibition of these human cells compared to *C. albicans* strain SC5314. While the published

 

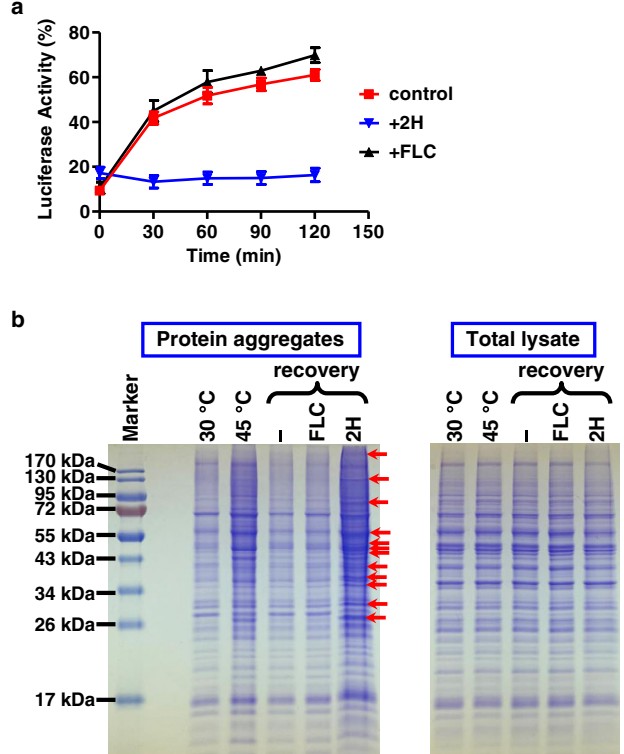

**Fig. 5 | 2H effectively inhibited both in vivo protein refolding and dismantling protein aggregates. a** In vivo refolding of luciferase in the *S. cerevisiae* strain *Msi3 sse1Δ*. A 45 °C treatment was used to denature the expressed luciferase. The luciferase activity before denaturation was set as 100%, and relative luciferase activities were plotted as a function of recovery time. The final concentration of 2H was 20 μM. FLC: fluconazole (100 μM). Control: without the addition of any compound. Data are presented as mean values +/- SEM (*n* = 6 for 2H and 4 for FLC-independent experiments). **b** Overall protein aggregation in vivo. Aggregated proteins (left panel) and total lysate (right panel) of the *S. cerevisiae* strain *Msi3 sse1Δ* were separated on SDS-PAGE. Cultures were grown at 30 °C. A treatment at 45 °C was applied to induce protein aggregation. The "30 °C" and "45 °C" samples were collected immediately before and after the 45 °C treatment, respectively. After the 45 °C treatment, the cultures were allowed to recover at 30 °C ("recovery" samples). 2H: 80 μM. FLC: fluconazole (100 μM). "–": without the addition of any compound. Protein markers are labeled on the left gel. Red arrows indicate selected unique or enriched bands in the sample treated with 2H compared to the 45 °C sample. Source data are provided as a Source Data file for all the panels.

human Hsp105 inhibitor had a strong inhibitory effect on HCT116[29], 2H showed a limited effect on these human cell lines, with estimated MIC$_{50}$ values all above 70 μM. This is consistent with the lower inhibitory effect of 2H on the chaperone activity of human Hsp105 compared to Msi3 (Fig. 3b).

To assess the potential of 2H as a general antifungal, we tested its effect on *C. glabrata*, a related *Candida* species, which is naturally more resistant to fluconazole than *C. albicans*[75]. Consistent with the high conservation between these two species of *Candida*, similar MIC$_{90}$ and MFC were observed for a common *Candida glabrata* strain 2001 (Table 1).

## 2H eliminates both in vivo protein folding and dismantling protein aggregates

Based on the inhibition of 2H on the chaperone activity of Msi3 in our biochemical analyzes above, we hypothesized that the antifungal effects of 2H result from a reduction in Msi3's in vivo chaperone activity. Msi3 is essential for many fundamental processes in maintaining proteostasis, such as protein folding and solubilizing protein aggregates[13,14,19,47–49,76,77]. To test this hypothesis, we took advantage of

the powerful and facile genetics available to *S. cerevisiae*. Closely related to *C. albicans*, *S. cerevisiae* is often used as a model organism to study fungal pathogens, despite being typically nonpathogenic[78]. Sse1 is the major Hsp110 in *S. cerevisiae*[58,59]. Sharing 63.4% sequence identity with Sse1, Msi3 can functionally substitute Sse1 in rescuing the temperature-sensitive phenotype of an *sse1* deletion strain (*sse1Δ*)[22,66]. Using a *S. cerevisiae* strain carrying *MSI3* and an *sse1* deletion (*MSI3 sse1Δ*), we have shown that 2H inhibits the growth and viability of *S. cerevisiae* in a manner similar to that observed for *C. albicans* (Table 1).

To analyze protein folding in vivo, we used firefly luciferase as a model substrate and expressed it in the *MSI3 sse1Δ* strain under the inducible promoter MET25[79]. We then inhibited further expression of luciferase by cycloheximide and denatured the expressed luciferase in vivo by treatment at 45 °C, a denaturing temperature for most proteins. As shown in Fig. 5a (at the time point zero), luciferase activity was reduced by ~90%, suggesting that most of the expressed luciferase was denatured. After shifting the cells back to 25 °C, the recovery temperature, luciferase activity was mostly restored over a period of 2 h, representing protein refolding after heat-denaturation (Fig. 5a). In the absence of heat denaturation, the luciferase activity remained stable (Fig. S11a), consistent with the lack of further expression due to the cycloheximide treatment. Strikingly, there was little increase in luciferase activity when 2H was included during the 25 °C recovery period (Fig. 5a), although the protein level of luciferase was comparable to the non-treatment control (Fig. S11b). This suggests that 2H effectively eliminated the in vivo protein refolding of luciferase, which is consistent with the strong inhibition of the in vitro chaperone activity shown in Fig. 3. The cells remained largely intact for every treatment (Fig. S12). In contrast, little impact on the refolding of luciferase was observed for fluconazole (Fig. 5a), consistent with its little to no impact on Msi3's holdase activity (Fig. S7b). Therefore, the inhibition of in vivo protein refolding is specific to 2H.

We analyzed the impact of 2H on dismantling protein aggregates in vivo using the *MSI3 sse1Δ* strain. To evaluate protein aggregation, we have isolated aggregated proteins using centrifugation and separated them on SDS-PAGE. We have reproduced the results that heat shock treatment at 45 °C results in increased protein aggregation compared to the no-heat shock control at 30 °C (Fig. 5b). The majority of the protein aggregates were solubilized after 90 min of recovery at 30 °C, corresponding to the in vivo activity of dismantling protein aggregates (Fig. 5b, lane "–"). In contrast, when 2H was included during the recovery phase, strong protein aggregation was observed while there was little change in the overall cellular proteins (Fig. 5b). Thus, 2H inhibits the in vivo activity of dismantling protein aggregates, consistent with its strong inhibition on the in vitro chaperone activity of Msi3. In contrast, little impact was observed for fluconazole (Fig. 5b, lane "FLC"), suggesting that the inhibition of dismantling protein aggregates is specific to 2H. Notably, the overall protein aggregation in the 2H-treated sample was even stronger than that induced by the 45 °C treatment. Besides enhanced intensity for several aggregated bands, multiple new bands were observed in the aggregates (labeled as red arrows in Fig. 5b), suggesting that 2H treatment results in a failure in overall protein folding during the recovery phase since proteins that fail to fold properly tend to aggregate[80,81]. This result further supports the inhibitory effect of 2H on protein refolding in vivo using luciferase as a model substrate observed above. These newly appeared and enhanced bands most likely represent endogenous substrates for Msi3's holdase activity, as they are uniquely enriched in protein aggregates in the sample treated with 2H when compared to the total lysate and the aggregated proteins at 45 °C. It is possible that the enhanced aggregation of some of these proteins could mainly be due to their high cellular abundance. To further confirm the inhibition of 2H on overall protein folding in vivo, we treated the *MSI3 sse1Δ* strain with 2H and analyzed protein aggregation. As shown in Fig. S13a,

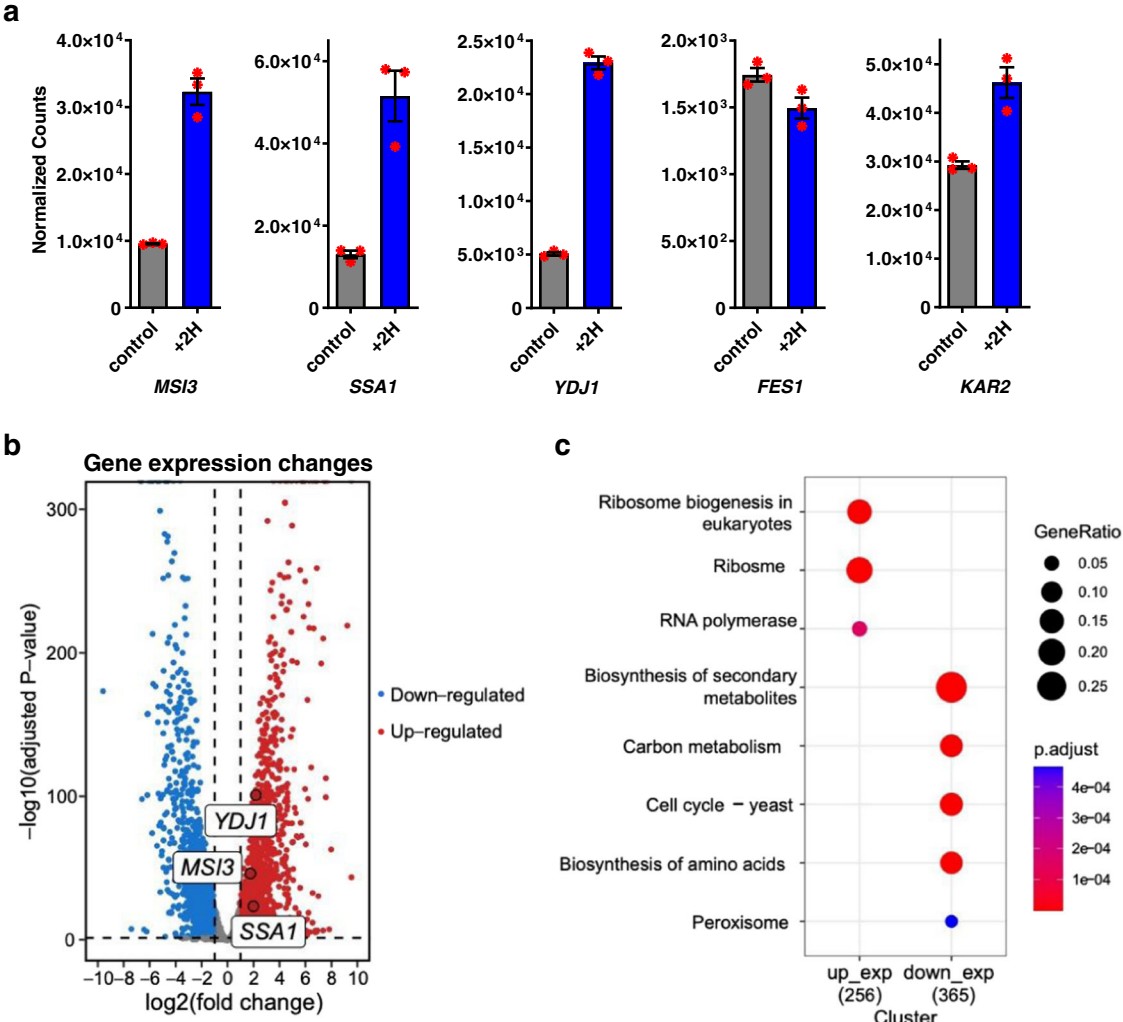

**Fig. 6 | The 2H-induced transcriptional changes in SC5314. a** The transcriptional changes of selected chaperones. Normalized counts of transcription are plotted. control: the samples without 2H treatment. The gene names are labeled under each plot. Source data are provided as a Source Data file. Data are presented as mean values +/- SEM (*n* = 3 independent experiments). Individual data points are shown in red. **b** The overall changes of gene expression upon 2H treatment. Fold of change was calculated using the ratios of normalized counts of the 2H-treated SC5314 samples over the non-treatment controls. DEseq2 was used to identify differentially expressed genes through a Wald test, employing a two-tailed p-value approach (adjusted using the Benjamini-Hochberg method). **c** The top pathways in SC5314 influenced by 2H treatment. The scales of gene ratio (GeneRatio) and adjusted P-value (p.adjust) are listed on the right. Up-regulated expression (up_exp) and down-regulated expression (down_exp) are plotted. A one-sided version of Fisher's exact test was performed, and the *p*-value was adjusted using the Benjamini-Hochberg method.

---

significantly enhanced protein aggregation with a pattern and intensity resembling that of the 2H treatment in Fig. 5b was observed, supporting the importance of Msi3's holdase activity in overall protein folding in vivo. Consistent with this observation, enhanced protein aggregation has been reported in *sse1Δ* strains[80,81]. Additionally, the impact of 2H on overall protein folding was reproduced in *C. albicans* SC5314 (Fig. S13b).

Taken together, 2H inhibits protein folding and proteostasis in both *S. cerevisiae* and *C. albicans*, which is consistent with its inhibition on growth and viability. This inhibition on proteostasis is most likely caused by the inhibitory effect of 2H on the holdase activity of Msi3, providing direct evidence for the crucial role of Msi3's holdase activity in maintaining proteostasis in vivo.

## 2H induces the expression of Msi3 and related chaperones in *C. albicans*

We have conducted a transcriptomic analysis on *C. albicans* strain SC5314 to evaluate the effect of 2H treatment on gene expressions. Interestingly, the transcription of Msi3 was induced by more than

3-fold upon 2H treatment (Fig. 6a, b, and Table S1), consistent with the inhibition of Msi3's activity by 2H shown above. Furthermore, several cytosolic chaperones functionally related to Msi3—including Ssa1 and Ydj1, the two partners of Msi3 in protein folding used in our in vitro refolding assay—were induced (around 4-fold each for Ssa1 and Ydj1). The transcription of Jjj3 and Caj1, two cytosolic Hsp40s, was also enhanced by more than 9-fold, partially because their transcription levels were relatively low in the absence of 2H treatment. Although the functions of Jjj3 and Caj1 are not as well characterized, they are likely co-chaperones for Ssa1 and Msi3, as their known functions overlap. HSF, the master transcription factor for heat shock proteins including Msi3, was also induced, albeit to a lesser extent. In contrast, no significant change was observed for several cytosolic chaperones, such as Ssb1, Ssz1, Zuo1, and TRiC/CCT (Table S1). This suggests that 2H treatment affects different cytosolic chaperones in distinct ways, which is consistent with the different functions and mechanisms among them[42,77]. Fes1 is another NEF in the cytosol for Ssa1 besides Msi3[20,82], but unlike Msi3, it has no holdase activity. Interestingly, there was limited impact on its transcription. In addition, neither Kar2 nor

---

Ssc1—the primary Hsp70s in ER and mitochondria, respectively—showed a significant change in transcription levels. Overall, these alterations in chaperone transcription are consistent with the unique inhibition of 2H on Msi3's holdase activity.

Aside from changes in chaperone expression, the top cellular processes impacted by 2H treatment are shown in Fig. 6c. The gene ratios, which represent the percentages of genes affected, range from 5% to 30% among these processes, with only three processes having gene ratios above 15%. The biosynthesis of secondary metabolites is the most affected process, with a gene ratio of approximately 30%; however, this process is not essential under normal conditions. Thus, no significant changes in transcription related to essential cellular processes are observed in response to 2H treatment, as indicated by the lack of significant alterations in gene ratio. Fluconazole treatment has been shown to induce transcriptional up-regulation of several genes in ergosterol biosynthesis, such as *ERG11* (the target of fluconazole), *ERG1*, *ERG3*, *ERG9*, and *CYB5*[83]. None of these genes showed significant up-regulation upon 2H treatment (Fig. S14a), consistent with the different cellular targets for 2H and fluconazole. Interestingly, a significant induction of several multidrug efflux pumps and transporters (such as *MDR1*, *CDR1*, and *CDR2*) was observed upon 2H treatment (Fig. S14b). This finding is consistent with the general function of these pumps and transporters in cellular detoxification of numerous antifungal compounds and toxic metabolites[84].

High osmolality (achieved in ways such as adding 1 M sorbitol to the growth medium) has been shown to suppress the temperature-sensitive phenotype caused by the loss of *SSE1* (i.e., *sse1Δ*) in *S. cerevisiae*[59]. This suppression has been attributed to the role of Sse1 in assisting Hsp90 in the cell wall stress pathway. However, the suppression by high osmolality is only partial for the *S. cerevisiae sse1Δ* strain used in our study (Fig. S15), suggesting that the degree of this suppression could be strain-specific. Including 1 M sorbitol in the medium showed little impact on either the MIC or MFC of 2H for SC5314 (Table 1), suggesting that suppressing the loss of Msi3's role in cell wall maintenance using high osmolality alone is not sufficient to support viability. Consistent with this result, the transcription of Hsp90 is not significantly impacted by 2H treatment (Table S1).

## Discussion

In this study, we have identified 2H as the first inhibitor for fungal Hsp110 Msi3. Importantly, 2H not only abolishes the chaperone activity of Msi3 through a novel mode, but also strongly inhibits the growth and viability of *C. albicans*. Thus, 2H is important for both understanding the molecular mechanisms of Hsp110s and developing a new class of antifungals. Additionally, this study provides direct support for fungal Hsp110s, such as Msi3, as important new targets for designing specific and efficient treatments against various fungal infections.

An exciting and unique feature of 2H is its ability to specifically abolish the holdase activity while leaving the NEF activity largely intact. This is the first reported data to show that the holdase activity is specifically compromised for an Hsp110. Although Hsp110s is an essential component of the Hsp70 chaperone machinery in many crucial processes in maintaining proteostasis[11,15,17–19,34,35,41–53], the molecular mechanism and exact role of Hsp110s are largely unclear. Mutational analyzes evidently support the importance of the NEF activity[61], but the function and involvement of the holdase activity are almost completely unknown and thus largely ignored. As the major NEF for cytosolic Hsp70s, Hsp110s are much larger than all other known classes of NEFs, none of which shows holdase activity like Hsp110s[19,20]. More importantly, Hsp110s cannot be functionally replaced by other classes of NEFs, such as Fes1, especially in the role of dismantling protein aggregation[19,47–49,76], implying the functional importance of the holdase activity. However, it has been challenging to dissect the function or mechanism of the holdase activity, primarily due to the lack of any available approach to disrupt this holdase activity without affecting the NEF activity. Thus, the strong inhibition of 2H on the Msi3-dependent refolding activity presented in this study has provided the first evidence to directly support that the holdase activity of Hsp110s is crucial for the folding activity of the Hsp70/Hsp110 chaperone machinery. Hence, this unique feature of 2H in inhibiting Msi3's chaperone activity suggests that it could serve as an invaluable chemical tool to explore the elusive molecular mechanism and in vivo function of Hsp110s.

In fact, we have demonstrated that the specific protein aggregates induced by 2H treatment are distinct from those induced heat shock, indicating that these unique aggregates most likely represent endogenous substrates for Msi3's holdase activity. Identifying these proteins using Mass Spectrometry would provide an unprecedented opportunity to characterize endogenous substrates for an Hsp110, allowing us for the first time to directly assess the functions and roles of the holdase activity of an Hsp110 in maintaining proteostasis in vivo. Because of the relatively high functional conservation among Hsp110s, the findings from the yeast system are likely applicable to other eukaryotes. Furthermore, the reduced protein folding and enhanced protein aggregation induced by 2H treatment indicate an overall disruption of proteostasis, the most probable underlying cause of the observed fungicidal effect of 2H. Thus, analyzing the unique protein aggregates induced by 2H treatment may also shed light on the contribution of specific cellular proteins and pathways compromised by the lack of Msi3's holdase activity to the fungicidal activity of 2H.

The location of 2H binding to Msi3 and its associated inhibition mechanism are of great interest. Our biochemical analyzes revealed that 2H reduces the peptide substrate binding affinity for Msi3 while leaving the NBD-SBD allosteric coupling largely intact. Since 2H binds Msi3 in the presence of a high ATP concentration, the observed reduction in the ATP binding affinity is most likely a secondary effect of the compromised peptide substrate binding. Although a direct analysis of the ATP binding affinity for the isolated NBD could resolve this issue, our expression tests indicated that the isolated NBD of Msi3 is unstable, consistent with a similar report on Sse1[16]. Therefore, we hypothesize that the binding site of 2H is likely near or at the substrate binding site in the SBD, and 2H inhibits the chaperone activity of Msi3 by reducing the affinity for substrate binding. This is consistent with the specific inhibition of 2H on the holdase activity of Msi3. However, since the substrate binding activity of Hsp110s, which is crucial for the elusive holdase activity, is ill-defined, predicting the binding site for 2H remains challenging. In fact, neither the substrate binding site nor substrate preferences is clear for Hsp110s. This is mainly because limited peptide substrates have been analyzed for Hsp110s, as the transient nature of peptide substrate binding to Hsp110s complicates substrate identification and structural characterization[36]. So far, no structure is available for any Hsp110s with a substrate bound, partially due to the lack of a substrate with both high-affinity binding and practical solubility. Thus, the identification of endogenous substrates for Msi3 through analyzing the specific protein aggregates induced by 2H treatment may provide insights into substrate binding properties and generate suitable substrates for structural and biochemical characterization.

Hsp110s are homologs of Hsp70s, and the substrate binding site and properties of Hsp70s are well established[85–87]. Despite this, the sequence conservation in the SBD between Hsp110s and Hsp70s is too low to be recognized as homologs (12-15% identity)[11,18,36,54], especially in the region around the known substrate binding site of Hsp70s. This makes the structure predictions based on homology unreliable. In fact, several mutational studies, including our own, suggest that the substrate binding site and mode of Hsp110s differ considerably from those of Hsp70s[36,52,56,88], consistent with their different substrate preferences[63,80,81,89–94]. It is not surprising that mutations targeting the analogous Hsp70 substrate binding site show little or limited impact on the substrate binding activity of Hsp110s[36,56,61]. Although a recent

in vitro biochemical study using such a mutant stated that the substrate binding by Sse1 is not obligated for its biological activities[56], this mutant still displayed a temperature-sensitive phenotype, consistent with the importance of the substrate binding activity in vivo. The lack of a significant defect in vitro assays for this mutant may reflect differences between in vivo and in vitro experimental conditions. A crystal or cryoEM structure of Msi3 in a complex with 2H will eventually reveal the binding site of 2H, which will be invaluable for characterizing the elusive substrate binding activity of Hsp110s.

As a promising lead compound for a new class of antifungals, 2H seems to have notable advantages over available antifungals. The three available antifungal drugs target two major pathways: sterol ergosterol (polyenes for binding directly and azoles for biosynthesis) and the cell wall (echinocandins). As a key player in the essential processes of protein folding and proteostasis, Msi3 represents a brand-new target. Thus, it is unlikely that there will be immediate clinical resistance to 2H, meaning that 2H is likely be effective against various *C. albicans* strains that are resistant to other antifungals. Indeed, our tests on four fluconazole-resistant strains support this hypothesis. Targeting proteostasis may therefore be an effective approach to overcome resistance.

There are various classes of molecular chaperones, each with distinct functions and mechanisms[42,77]. Consistent with our study, inhibitors of Hsp90 molecular chaperones have demonstrated potential for inhibiting the growth of *C. albicans*[95,96]. Interestingly, Hsp110s have been identified as cochaperones for Hsp90s[17,43]. As antifungal targets, it seems Hsp110s have key advantages over Hsp90s. Hsp90s are essential and highly conserved, making it challenging to achieve selectivity against fungal pathogens. Three Hsp110s have been reported in humans: Hsp105, Apg-1, and Apg-2. Hsp105 serves as the primary member and is expressed in nearly all the tissues examined, while Apg-1 and Apg-2 are typically expressed in the testes. Knockout experiments in mice suggest that none of these human Hsp110s are necessary for normal development and growth[23–25]. In contrast, Hsp110s, such as Msi3, are crucial for fungi. Moreover, the conservation among Hsp110s is relatively low (Fig. S2 and S16). In addition, with extra sequences at the C-terminus, human Hsp110s (-110 kDa) are much larger than fungal Hsp110s (-80 kDa), suggesting that they may utilize different mechanisms[11,18,88]. Indeed, unlike Msi3, Hsp105 is unable to substitute for Sse1 in supporting growth (Fig. S8a). Taken together, Hsp110s may have higher selectivity as an antifungal target than Hsp90s. Recently, inhibitors selectively targeting fungal Hsp90s were reported, despite the high sequence conservation[95], which bolsters confidence for targeting Hsp110s to treat fungal infections.

The potential binding site of 2H on the SBD could provide an advantage for 2H as a selective inhibitor for Msi3 over human Hsp110s. Based on the sequence alignment between human Hsp110s and Msi3 (Fig. S16), the conservation in the SBD is especially low, even when considering the already low overall sequence conservation. Moreover, the SBD of human Hsp110s is significantly larger than that of fungal Hsp110s, and human Hsp110s may use different mechanisms and sites for substrate binding[88]. On the other hand, the NBD shows higher sequence and structure conservation[29,54,60,61], which may be due to its conserved role in binding ATP. Thus, the nucleotide-binding site may not be an appropriate target for specific inhibitors.

Based on the high conservation among fungal Hsp110s (Figs. S2), 2H has the potential to function as a broad-spectrum fungicidal. Besides *C. albicans*, the most common pathogen for fungal infections, multiple *Candida* species can cause serious infections in humans, including *C. glabrata*, *C. parapsilosis*, *C. tropicalis*, *C. krusei*, and *C. auris*. Hypothetical Hsp110 homologs have been reported in some of these *Candida* species. The sequence conservation among these fungal Hsp110s is high (Fig. S2). Thus, Hsp110s are most likely essential for these *Candida* species, and 2H may be effective against candidiasis

caused by various *Candida* species. Our observation of 2H's fungicidal effect on *C. glabrata* supports this hypothesis.

A limitation of 2H is its moderate solubility in aqueous solutions ($\leq$100 $\mu$M, 43.7 mg/L), although its solubility in organic solvents such as DMSO can reach more than 20 mM. To increase the efficacy of 2H for inhibiting *C. albicans*, the next step in our effort is to improve its solubility by designing and synthesizing chemical analogs. In addition, the structural basis of 2H binding to Msi3 will provide a solid foundation for designing next-generation inhibitors with increased binding affinity and enhanced selectivity for Msi3. Such analogs of 2H improved in all these aspects have the potential to aid the development of effective antifungals with enhanced potency.

## Methods

### Protein expression and purification

The expression and purification of Msi, Msi3-I164D, Ssa1, Ydj1, Sse1, human Hsp70 (hHsp70), and firefly luciferase have followed published protocols[54,65,66,97–99]. Human Hsp105 and HDJ2 were expressed and purified in a similar way as that of Msi3. Except for Ssa1, the ORFs of these proteins were amplified and cloned into the pSMT3 vector to express as an Smt3 fusion protein with a His6 tag at the N-terminus. To increase the Ulp1 digestion efficiency during purification (see below), a linker (sequence: GSDS) was inserted between Hsp105 and Smt3. The pSMT3 vector was a generous gift from Dr. Christopher Lima (the Sloan Kettering Institute)[100]. The Msi3 ORF was amplified from a genomic DNA preparation of *Candida albicans* (generously provided by Dr. Ronda Rolfes, Georgetown University). The I164D mutation was introduced into Msi3 using mutagenic PCR. For Msi3, Hsp105, hHsp70, and HDJ2, the expression was carried out at 18 °C for 6-8 hours after transforming Novagen's Rosetta2(DE3)pLysS strain (MilliporeSigma). Ydj1, Sse1, and firefly luciferase were expressed in BL21(DE3) at 30 °C for 6 hours. All the Smt3 fusion proteins were first purified on a HisTrap column (GE Healthcare Life Sciences) using buffers containing 25 mM Hepes-NaOH, pH 7.5, 300 mM NaCl, 10% glycerol, and 1 mM TCEP. The eluted fractions containing the Smt3 fusion proteins were treated with Ulp1 protease to cleave off the Smt3 tag. After the Smt3 tag was removed on a second HisTrap column, the resulting proteins were further purified on a HiTrap Q column. The peak fractions were concentrated to >10 mg/ml in a buffer containing 25 mM Hepes-KOH, pH 7.5, 50 mM KCl, and 1 mM DTT.

The *Pichia pastoris* strain GS115 was used to express Ssa1 protein (a generous gift from Dr. Johannes Buchner at the Technische Universität München)[97]. After induction in YP medium contacting 0.5% methanol at 30 °C for 24 h, the expressed Ssa1 protein was first purified on a HiTrap Q column using buffers containing 25 mM Hepes-NaOH, pH 7.5, 1 mM EDTA and 1 mM DTT. The fractions containing Ssa1 were pooled, $(NH_4)_2SO_4$ was added to a final concentration of 1.5 M and then purified on a butyl-Sepharose column. Before mixing with a 3 ml ATP agarose resins (Sigma), the Ssa1 protein was dialyzed in a buffer containing 25 mM Hepes-NaOH, pH7.5, 150 mM NaCl, 10 mM Mg(OAc)$_2$ and 1 mM DTT. Then, the Ssa1 protein was eluted with 5 mM ATP. After extensive dialysis to remove ATP, the purified Ssa1 protein was concentrated to more than 10 mg/ml.

After transforming the pET28-ULP1 plasmid into BL21(DE3), the expression of Ulp1 was carried out at 30 °C for 5-6 hours. Ulp1 was purified on a HisTrap column using 2xPBS buffer. After dialyzed in a buffer containing 25 mM Hepes-NaOH, pH7.5, 300 mM NaCl, and 10% glycerol, the purified Ulp1 protein was concentrated to >10 mg/ml.

All the purified proteins were aliquoted, flash-frozen in liquid nitrogen, and stored in −80 °C freezers.

### Compound preparation and storage

Stock solutions in DMSO were prepared for 2H (20 mM), Riociguat (20 mM), Compound C (10 mM), and fluconazole (100 mM). Before stored in a −80 °C freezer, all stock solutions were aliquoted and flash-

frozen in liquid nitrogen. Each tube of compounds was thawed only once and used within several hours after thawing. In all assays, the concentration of each compound was kept below its solubility limit in aqueous solution.

## Fluorescence polarization assay for determining ATP binding

A fluorescence-labeled ATP, N6-(6-amino)hexyl-ATP-5-FAM (ATP-FAM), was purchased from Jena Bioscience (Germany) for testing ATP binding to Msi3, Sse1, Hsp105, and Ssa1 using a fluorescence polarization assay[56,65,66]. For the compound screen, Msi3 was diluted to 0.1 μM using Buffer A (25 mM Hepes-KOH, pH 7.5, 150 mM KCl, 10 mM Mg(OAc)$_2$, 10% glycerol, and 1 mM DTT). Then, each compound was added to a final concentration of 100 μM and incubated at the room temperature for 1 hour. After adding ATP-FAM at a final concentration of 20 nM, the reactions were incubated for 1 h at the room temperature to allow ATP-FAM to bind to Msi3. The fluorescence polarization measurements were performed on a Beacon 2000 instrument (Invitrogen), and the reaction without any compounds was used as a positive control.

To determine the effect of compounds on the binding affinities of ATP-FAM for Msi3, Sse1, Hsp105, and Ssa1 proteins, serial dilutions of each protein were incubated with each compound at the indicated final concentrations for 1 hour at the room temperature. Then, ATP-FAM was added to a final concentration of 20 nM, and the reactions were incubated for 1 hour at room temperature. Fluorescence polarization was measured for each reaction and plotted using GraphPad Prism software to deduce dissociation constants ($K_d$) after fitting to a one-site binding equation.

## 2H fluorescence spectrum for analyzing 2H binding to Msi3

To assay 2H binding to Msi3, we scanned the fluorescence spectra of 2H in the presence and absence of Msi3 with the excitation wavelength set at 308 nm. 2H was kept at a final concentration of 5 μM for all the scans. For the Msi3 protein with 2H treatment, 10 μM Msi3 was incubated with 2H in buffer A for 1 h at room temperature before scanning the emission spectra. For reactions with ATP, a 2 mM final concentration of ATP was included. When relative fluorescence was used, peak intensity of each spectrum was set at 1.

## Determining peptide substrate-binding affinity using fluorescence polarization

The TRP2-181 peptide labeled with a fluorescein at the N-terminus was used to determine the substrate binding affinity of Msi3 based on a previously published protocol[65]. In brief, to determine the effect of 2H on the binding affinity of Msi3 for the TRP2-181 peptide, serial dilutions of Msi3 were prepared in buffer A, and 2H was added to the indicated final concentrations. After incubating at room temperature for 1 hour, the TRP2-181 peptide was added to a final concentration of 20 nM. After mixing well, the reaction was further incubated at room temperature for 1 h to allow the binding to reach equilibrium. Then fluorescence polarizations were recorded and fitted to a one-site binding equation to calculate dissociation constants ($K_d$) using GraphPad Prism. An almost identical protocol was used to determine the binding of the TRP2-181 peptide to Hsp105 and the NR peptide to Ssa1.

For the competition analysis, Msi3 at the indicated concentration was first incubated with the TRP2-181 peptide for 30 min to allow binding to reach equilibrium. After polarization was measured to confirm the binding, 2H was added to a final concentration of 100 μM, and polarization readings were tracked over time to monitor the release of the TRP2-181 peptide. The addition of an equal amount of the solvent DMSO was used as a negative control.

## Limited trypsin digest for analyzing ATP-induced allosteric coupling in Msi3

This assay was performed using a previously published protocol[54,65,66]. In brief, Msi3 was diluted to 1 mg/ml in buffer A and incubated with 2H

for 1 hour at room temperature. Then, trypsin was added to a final concentration of 1.67 μg/ml and incubated at 25 °C for 30 min. The digest reactions were terminated by adding PMSF to a final concentration of 1 mM. After running SDS-PAGE gels, the digest patterns were visualized via staining with Coomassie blue.

## NEF activity assay for Msi3

The assay was performed based on a previously published procedure[65,66]. ATP-FAM (20 nM final concentration) was incubated with Ssa1 protein (1 μM final concentration) on ice for 1 h to form a Ssa1-ATP-FAM complex in buffer A. Meanwhile, in separate tubes, Msi3 was diluted to 10 μM and incubated with 2H at a final concentration of 100 μM for 2 hours at room temperature. Afterward, 12 μl of this 2H-treated Msi3 was mixed rapidly with 108 μl of the preformed Ssa1-ATP-FAM complex at room temperature, and fluorescence polarization readings were recorded over time using a Beacon 2000. Regular ATP (final concentration: 50 μM) was included in the indicated reactions. Msi3 without 2H treatment was used as a control.

## Native gel analysis on the complex formation between Msi3 and Ssa1

Purified Msi3 and Ssa1 proteins were diluted using buffer D (25 mM Hepes-KOH, pH 7.5, 50 mM KCl, 10 mM Mg(OAc)$_2$, and 1 mM DTT). 2H (at a final concentration of 100 μM) was incubated with diluted Msi3 for 1 hour on ice. When forming the Msi3-Ssa1 complex, Msi3 (5 μM) and Ssa1 (10 μM) were mixed with ATP at a final concentration of 2 mM and incubated on ice for 45 minutes. After adding the sample buffer, all the samples were applied to 10% native gels (run at 120 V for 3 hours on ice). The native gels were stained with Coomassie Blue to visualize the bands.

## Preventing protein aggregation (i.e., holdase) assay using OD$_{320}$

When purified firefly luciferase was used as a substrate, the assay was carried out using a published protocol[34,65,66]. Before performing the assay, 90 μM Msi3 or Msi3-I164D was first treated with 2H (100 μM) at room temperature for 2 h in buffer B (25 mM Hepes-KOH, pH 7.5, 150 mM KCl, 10 mM Mg(OAc)$_2$, 3 mM ATP, and 1 mM DTT). Then, the 2H-treated Msi3 was mixed with luciferase in buffer B. The final concentrations of luciferase, Msi3, and 2H were 750 nM, 9 μM and 10 μM, respectively. The reactions (1 ml for each reaction) were started by heating in a 42 °C water bath, and UV absorbance at 320 nm was monitored over time. Luciferase alone was used as a negative control, and luciferase with the addition of Msi3 as a positive control.

For the assay using Ulp1 as a substrate, the methods described above were repeated using purified Ulp1 protein in place of luciferase. The final concentrations of Ulp1, Msi3, and 2H are 2.5, 10, and 20 μM, respectively.

## Holdase assay for IC$_{50}$ determination

The assay was carried out in a similar way as published previously for human Hsp110[29]. An 11-min incubation at 42 °C was applied on purified luciferase (4 nM) to induce denaturation, which results in aggregation in the absence of chaperones. Inclusion of Msi3 (0.5 μM) during this incubation prevents the denatured luciferase from aggregating. Afterward, Ssa1 and Ydj1 were added to a final concentration of 3 μM each to refold the denatured luciferase while aggregated luciferase failed to refold. After incubating at room temperature for 30 min, the activity of luciferase was measured in a luminometer (Berthold LB9507) by mixing 2 μl of each reaction mixture with 50 μl of luciferase substrate (Promega). Buffer C (25 mM Hepes-KOH, pH 7.5, 100 mM KCl, 10 mM Mg(OAc)$_2$, 3 mM ATP, and 1 mM DTT) was used for this assay. To determine the IC$_{50}$ for 2H, a serial dilution of 2H was incubated with Msi3 (5 μM) for 1 hour at room temperature before incubating with luciferase at 42 °C. Relative activities were calculated by setting the luciferase activities in the samples with the untreated Msi3

as 100%. As compound controls, Msi3 was treated with either fluconazole, Riociguat, or Compound C in the same way as 2H.

An almost identical method was used for human Hsp105. To achieve a strong activity in preventing the aggregation of luciferase, the final concentrations of Hsp105 were 4 μM. hHsp70 and HDJ2 were used in place of Ssa1 and Ydj1.

### In vitro refolding assay using purified proteins

The in vitro refolding assays using the Ssa1 and human Hsp70 chaperone systems with purified firefly luciferase as a substrate were carried out based on a published protocol with modifications[14,61,65,66,98]. For the refolding assay using the Ssa1 chaperone system, Ssa1, Msi3, and Ydj1 were used as Hsp70, Hsp110, and Hsp40 chaperones, respectively. First, 60 μM Msi3 was treated with 100 μM 2H in buffer C for 1 hour at room temperature. Then, a refolding reaction mixture containing 3 μM Ssa1, 4 μM Ydj1, and 2 μM 2H-treated Msi3 in buffer C was prepared. After luciferase (11 nM) was denatured by incubating at 42 °C for 15 min in the presence of 3 μM Ssa1, the refolding reaction was started by diluting the denatured luciferase into the refolding reaction mixture. At the indicated time, the activity of luciferase was measured using a luminometer (Berthold LB9507) by mixing 2 μl of refolding reactions with 50 μl of luciferase substrate (Promega). Msi3 with no treatment was used as a control. Relative luciferase activities were calculated by setting the activity of the unheated luciferase as 100%.

The human Hsp70 chaperone system used includes hHsp70, Hsp105, and HDJ2. For the refolding assays, we carried out an almost identical protocol as that of the Ssa1 chaperone system described above. The final concentrations of hHsp70, Hsp105, and HDJ2 were 3, 1, and 3 μM, respectively.

### Antifungal susceptibility and viability testing on yeast strains

MIC and MFC were determined according to the Clinical and Laboratory Standards Institute for yeasts (CLSI, M27). The WT *C. albicans* strain SC5314 was obtained from Dr. Ronda Rolfes and the fluconazole-resistant strains FH5, 12-99, JS14 and JS15 from Dr. Theodore White. We purchased the *C. glabrata* strain 2001 and *S. cerevisiae* strain *YPL106C BY4742 (MATα his3Δ1 leu2Δ0 lys2Δ0 ura3Δ0 trp1Δ1 sse1Δ)* from ATCC. To accommodate all the pathogenic strains, a yeast synthetic medium YNB was used for the tests since some of the strains were reported to have poor growth in RPMI medium[74]. Before stored in the −80 °C freezer, all these strains were grown in Sabouraud medium, resuspended in 35% glycerol, aliquoted, and flash frozen in liquid nitrogen.

A frozen aliquot was thawed and inoculated in 5 ml Sabouraud medium. After growing at 30 °C for 1 overnight, the culture was centrifuged at 1200 g for 5 min at room temperature and washed twice with PBS. The inoculum was prepared by diluting the washed culture into YNB medium to a final $OD_{600}$ of 0.0005 or $5 \times 10^3 - 1 \times 10^4$ colony forming unit (CFU) per ml. The inoculum was aliquoted into a sterile 96-well culture plate. Serial dilutions of compounds (such as 2H, fluconazole, Riociguat, or Compound C) were prepared and added into the aliquoted inoculum. The final volume is 100 μl for each well. The plate grew at 37 °C with shaking at 250 rpm. The growth was measured by absorbance at 600 nm in a spectrophotometric plate reader. For SC5314, 12-99, JS14, JS15, and 2001, a growth of 24 h was used. FH5 grows slower and readings were measured after 48 hours. Wells with medium only were used as blanks and wells with only inoculum were used as growth controls. As a viability test to determine MFC, after measurement with a spectrophotometric plate reader, a 100 μl aliquot was removed from each well that showed no obvious reading (i.e., growth), diluted with 100 μl PBS, and plated on to a Sabouraud agar plate. After growing at 30 °C for 1–2 days, colonies were counted. Chequerboard (or checkerboard) assay was carried out to determine the fractional inhibitory concentration index (FICI) according to the published protocol[101,102].

After transforming the *YPL106C BY4742 sse1Δ* strain with the pRS313-MSI3 plasmid[65,66], we obtained the strain *MSI3 sse1Δ* used for our assay[65,66]. The growth and viability tests were carried out in a yeast synthetic dropout medium without histidine (His dropout) to maintain the pRS313-MSI3 plasmid[65,66].

The expression of Hsp105 was assayed using Western blot with an anti-HSPH1 antibody (R&D System, cat # AF4029) using 1:2,000 dilution. The secondary antibody is Rabbit anti-Goat Ig Secondary Antibody, HRP (Invitrogen, cat # SA5-10314) at 1:40,000 dilution.

### Biofilm formation in vitro

The assay was performed according to a published protocol[103]. SC5314 was inoculated in Sabouraud medium and grown at 30 °C for 1 overnight. The culture was centrifuged at 1200 g for 5 min at the room temperature and washed twice with PBS. After resuspended in YNB medium, the $OD_{600}$ was adjusted to 0.005 ($5 \times 10^4 - 1 \times 10^5$ CFU/ml). A sterile 96-well tissue culture plate was coated with 10% fetal bovine serum for 2 hours at room temperature. After washing each well twice with 200 μl PBS, the inoculum was aliquoted into the 96-well plate. A serial dilution of 2H was prepared and added to wells with inoculum. The plate was incubated at 37 °C without shaking. After 24 hours, the plate was washed twice with PBS to remove cells that did not adhere. Adherent cells (i.e., biofilm) were assessed by both measuring absorbance at 600 nm in a spectrophotometric plate reader and examining with light microscopy.

### Cell cytotoxicity assay

Human lung fibroblast HLF-1 cell line, human hepatic stellate LX-2 cell line and human colorectal cancer HCT116 cell line were purchased from ATCC (Manassas, VA). HLF-1 cells were cultured in DMEM-F12K medium (Gibco, Carlsbad, CA), and LX-2 and HCT116 cells were cultured in DMEM medium (Gibco, Carlsbad, CA). All the cells were grown at 37 °C and 5% $CO_2$ under humidified conditions with media supplemented with 10% fetal bovine serum (Gibco, Carlsbad, CA), 100 mg/mL penicillin G, and 100 mg/mL streptomycin (Life Science, P1400). Cell cytotoxicity was determined using a CCK-8 assay. In brief, HLF-1, LX-2 and HCT116 cells were seeded separately into 96-well plates with $3 \times 10^3$ cells/well at a final volume of 100 μl. After 12 h growth, cells were treated with the indicated concentrations of 2H for 48 h. Then, fresh CCK-8 (10 μl, 5 mg/mL, Biosharp) was added to each well and incubated at 37 °C for 2 h. Finally, the spectrophotometric absorbance of each well was measured using a microplate reader at a wavelength of 450 nm.

### In vivo luciferase refolding assay in yeast

The assay was carried out based on the protocol published by the Morano group[79]. The p426MET25-FFL-GFP plasmid that carries the firefly luciferase gene (FFL) for expression in *S. cerevisiae* as the model substrate for refolding was a generous gift from Dr. Kevin Morano. This plasmid was transformed into the *S. cerevisiae* strain *MSI3 sse1Δ*[65,66]. Fresh transformants were grown overnight at 30 °C in a yeast synthetic dropout medium without histidine and uracil. On the second day, the culture was diluted to 0.6 $OD_{600}$ and continued to grow at 30 °C for 2–3 h until it reached 0.8 $OD_{600}$. Then, the cells were pelleted by centrifugation at 3000 g for 3 min and washed twice with sterile water. To induce the expression of FFL, the yeast cells were resuspended in a yeast synthetic dropout medium without histidine, uracil, and methionine. After induction for 1 hour at 30 °C, cycloheximide was added to a final concentration of 100 μg/ml to inhibit further protein expression. To denature FFL, the yeast cells were heated in a 45 °C water bath for 30 min. Following the addition of 2H (20 μM) or fluconazole (100 μM), refolding was carried out at 25 °C. At the indicated time points, 50 μl samples were taken from each culture and mixed with 50 μl of D-luciferin (455 μg/ml final concentration). The luciferase activity was measured using a

luminometer (Berthold LB9507). Cultures without heat shock denaturation were used as controls. The luciferase activity before heat shock denaturation was set as 100%.

The protein levels of FFL were assayed using Western blot with an anti-luciferase antibody (Rockland, cat # 200-103-150 S) using 1:50,000 dilution. To analyze the integrity of the cells, cultures were examined using a Zeiss Spinning Disc Confocal Microscope (Objective: 100X Oil; Reflector: DIC-TL).

### In vivo protein aggregation assay

To assess the effect of 2H on dismantling protein aggregation and protein folding in vivo, the published protocols for analyzing protein aggregation were used[80,81]. For dismantling protein aggregation in vivo, a single colony of the *S. cerevisiae* strain *MSl3 sse1Δ* was inoculated into a yeast synthetic dropout medium without histidine. After growing one overnight, the culture was diluted to 0.2 $OD_{600}$ and continued to grow for another 3–4 hours to reach 0.4–0.5 $OD_{600}$. A 30-min heat-shock incubation at 45 °C was applied. Afterward, the culture was chilled briefly at 4 °C and allowed to recover at 30 °C for 90 min. 2H or fluconazole was added to a final concentration of 80 μM immediately before the 90-min recovery. Samples were taken immediately before and after heat-shock incubation as well as after a 90-min recovery. Each sample was centrifuged at 1200 g for 5 min to pellet the cells and flash frozen in liquid nitrogen. The cell pellets were thawed and resuspended in a cold PBS buffer with the addition of 1 mM EDTA, 2 mM DTT and 0.1% Tween 20. French Press was applied to break the cells. Intact cells and cell debris were removed by a centrifugation at 5000 g for 10 min. Protein concentrations of the cell lysates were estimated by $OD_{280}$ using NanoDrop. After adjusting the concentrations to a similar level, the cell lysates were subjected to an ultracentrifugation at 20,000 g for 20 min. The supernatant was removed, and the pellets were washed with a cold PBS buffer containing 1 mM EDTA, 2 mM DTT, and 2% NP40. After a second centrifugation at 20,000 g for 20 min, the resulting pellets were dissolved in lithium dodecyl sulfate (LDS) sample buffer. The total cell lysate and pellet for each treatment were loaded onto an SDS-PAGE. Equivalent amount of each sample was adjusted based on the $OD_{280}$ measurement. The same protocol without the 45 °C heat-shock was carried out to examine protein aggregation in vivo to analyze the effect of 2H on protein folding. A similar protocol was used to evaluate the effect of 2H (100 μM) on in vivo protein aggregation in *C. albicans* strain SC5314.

### Transcriptomic analysis

*C. albicans* strain SC5314 was inoculated in Sabouraud medium and grew at 30 °C for overnight. After centrifugation at 1200 g for 5 minutes, the pellet was washed twice with PBS, and re-suspended in YNB medium to an $OD_{600} = 0.2$. 2H was added to a final concentration of 100 μM and the culture was grown at 37 °C for 30 minutes. A culture with addition of DMSO was used as a control. Cells were pelleted by a centrifugation at 1200 g for 5 min and washed twice with PBS. Afterward, the cell pellet was flash frozen in liquid nitrogen. Triplicates were prepared for each sample. Total RNA was extracted from each sample using a TRNzol Universal Reagent (TianGen Biotechnology, Beijing, China). The purity and concentration of extracted RNA were estimated using NanoDrop 2000 (NanoDrop Technologies), and the integrity of the total RNA was analyzed using Agilent 2100 bioanalyzer (Agilent Technologies). mRNA was enriched from the total RNA using oligo (dT) magnetic beads, then digested into short pieces using the fragmentation buffer at 94°C for 5 min. Using mRNA fragments as templates, the first strand of cDNA was synthesized through random hexamer primers, followed by the second-strand of cDNA synthesis using DNA polymerase I and RNase H. The products were amplified by PCR and further purified by QIAquick PCR purification kit to create a cDNA library. The libraries were sequenced using the Illumina HiSeq

2500 platform (Illumina) at Novogene Bioinformatics Technology Co., Ltd (Beijing, China).

The RNA-seq data were aligned to the Ensembl *Candida albicans* annotation information using HISAT2 (version 2.0.5), and the quantification was performed with featureCounts (version 1.5.0–p3). The R package DEseq2 (version 1.38.3)[104] was then employed to identify differentially expressed genes between the control and 2H-treated groups, defined as genes with adjusted *p*-value < 0.05 and fold change >2. The R package Clusterprofiler[105] was used for pathway analysis of the differentially expressed genes, based on the definition from Kyoto Encyclopedia of Genes and Genomes (KEGG) database. Pathways with *p*-value < 0.01 and *q*-value < 0.05 were considered enriched.

### Statistics and reproducibility

All experiments were performed more than three times independently with similar results.

### Reporting summary

Further information on research design is available in the Nature Portfolio Reporting Summary linked to this article.

## Data availability

The RNA-seq data for the transcriptomic analysis generated in this study have been deposited in the Gene Expression Omnibus (GEO) under the accession code GSE226137. All other data described in the manuscript are contained within the manuscript. Source data are provided with this paper.

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

## Acknowledgements
We are grateful for the generous help from: Dr. Ronda J. Rolfes (Georgetown University) for the wild-type *C. albicans* strain SC5314 and various protocols of testing *C. albicans*, Dr. Kevin Morano (University of Texas Health Science Center at Houston) for the *p426MET25-FFL-GFP* plasmid, Dr. Theodore White (University of Missouri) for the fluconazole-resistant *C. albicans* strains (FH5, 12-99, JS14 and JS15), Dr. Dong Sun for plate reader, Dr. Carlos Escalante for discussion, and Dr. Kimberly Jefferson for French Press. We thank Drs. Shihao Song and Yinyue Deng (Zhongshan University) for their assistance with various analyzes, Dr. Julia Hotinger (Virginia Commonwealth University, School of Pharmacy) for assisting us with the Z-factor determination and discussion, and Ms. Ryan Wen and Tiffany Zhao for technical support and discussion, and Dr. Yaping Wang (Department of Internal Medicine, Virginia Commonwealth University) and Dr. Junchao Shi (Division of Biomedical Sciences, University of California, Riverside) for their help in analyzing transcriptomic analysis data and figure preparation. Microscopy was performed at the VCU-Department of Anatomy & Neurobiology Microscopy Facility (supported, in part, with funding from the NIH-NCl Cancer center support Grant P30 CA016059). This work was supported by NIH (R01GM098592 to Qinglian Liu), the Virginia Commonwealth University (VETAR and Bridge awards to Qinglian Liu), Virginia Commonwealth University CTSA (UL1TR002649 from the National Center for Advancing Translational Sciences) and the CCTR Endowment Fund of Virginia Commonwealth University (CCTR Award to Qinglian Liu), Shenzhen Bay Laboratory (#S211101001-3 and 2021CX02Y082 to Lei Zhou), and National Natural Science Foundation of China (#32171150 to Lei Zhou). We tried to include as many relevant references as possible and apologize if important citations were inadvertently omitted.

## Author contributions
Qinglian Liu, C.S., L.H., and L.Z. designed the study and experiments. L.H. and Q.Li synthesized the compound library and L.H., C.S, and A.E.M identified 2H after screening compound libraries. C.S., L.H., J.M.K., J.H., X.F., H.L., Qingdai Liu performed all the experiments. Qinglian Liu and C.S. analyzed the data and wrote the manuscript. All authors discussed the results, read the manuscript, and made revisions.

## Competing interests
The authors declare no competing interests.
