## [Peer Review File · Nature Communications]

A first-in-class inhibitor of Hsp110 molecular chaperones of pathogenic fungiReviewer #1 (Remarks to the Author):

In this work, Hu et al report an inhibitor of *Candida albicans* HSP110 (Msi3) that might be used for the treatment of fungal infections. The growth, viability and pathogenicity of *C. albicans* is inhibited by this drug, which they call 2h. Although the paper is potentially interesting there are important concerns. The results shown are insufficient to support the important conclusions drawn by the authors.

Specific comments:

1. There is a big problem concerning the controls used to demonstrate the specificity of 2h. First, in all figures another chemical molecule should be included as a negative control. Authors can use one obtained from their initial screening with no affinity for Msi3 (Fig 1A). Second, a more relevant control than DNAK (a HSP70 protein from *E. coli*) should be used. For instance, HSP70 from *C. albicans* and other HSP110 proteins such as human HSP110 (it is very important to demonstrate that 2h does not affect human HSP110 if the authors want to conclude that 2h does not affect human cells).
2. To demonstrate the role of 2h in the cells' growth and pathogenesis, a transcriptomic analysis showing that genes involved in these two processes are affected by 2h is needed. Authors could also use a proteomic approach to study how 2h affects the interactome of Msi3. To say that the effect of 2h is through protein homeostasis (supported only by a single luciferase assay) is not enough. Many chaperones have an effect in proteins homeostasis and there is a lot of redundancy in this function essential for cell viability. In this case, this can be just a consequence of blocking Msi3, but not necessary the cause of the effect of 2h on cell growth and/or pathogenesis. Deciphering the effect of 2h on Msi3s' interactome would also help select a relevant Msi3 substrate (more relevant than the TRP2 peptide).
3. Too few and too unclear results are shown to be able to conclude that the 2h molecule affects HSP110s' holdase activity without affecting its' NEF activity. For instance, in Figure 2C, although small, there is a difference between Msi3+ATP and Msi3+ATP+2h. And in Fig 3 B-D, I do not understand the choice of the DNAK chaperone machinery, a system that does not involve an HSP110 protein. Why should 2h have an effect? Why not use the HSP70 chaperone cycle (which uses HSP110) to study the effect of 2h? Does 2h bind to GrpE?
Authors should use mutants of Msi3 (with no NEF or holdase activity) to prove what they are claiming.
4. Fig 4: First of all, it seems that the authors are using a strain that is resistant to fluconazole. It has hardly any effect (the MCI50 for fluconazole seems to be over 50). No comparison between the two molecules is therefore possible. Authors could study both a resistant and a sensitive strain and the MCI80 should be presented.
5. Concerning the synergistic effect, the methodology used does not comply with the standards (Clinical Laboratory Standards Institute (CLSI) or European Committee on Antimicrobial Susceptibility Testing (EUCAST)). The interpretation of the results must take into account the fractional inhibitory concentration index (FICI).
6. Tolerance and resistance are not equivalent. Msi3 has been shown to be involved in tolerance (Nagao et al 2012). In this case, an impact of Msi3 on calcineurin signaling has been reported. Could this mechanism be involved in this synergy (if synergy there is)? An effect of 2h on protein (re) folding is proposed. Could 2h impact the target of fluconazole? The fluconazole efflux pump for example?
7. In Figure 4B, why use the colorectal cancer cell line HCT116? Authors cannot conclude just from the data shown in Fig 4B that 2h does not affect human cell growth. Other human cell lines, including non-cancerous cells, should be tested and shown in a different graph.
8. In fig 5, to study the fungicidal effect of 2h, MIC should be replaced by MFC. Furthermore, the effect of 2h combined to fluconazole should be included, using both a fluconazole resistant and a sensitive strain. And again, like in all the figures, a chemical compound that does not bind to Msi3 should be included as a negative control.
9. Inhibition of pathogenicity is claimed based on the ability of *C. albicans* to form filaments. Since this is an important conclusion, other tests should be performed, for

example a biofilm formation inhibition test and/or an adhesion inhibition test.

10. In Fig 6, rescue experiments could be included by addition of recombinant Msi3 (and human HSP110 as a negative control, if the authors prove in Fig 1 that 2h does not bind/affect human HSP110).

11. Finally, in the discussion it is mentioned that 2h has solubility issues. However, nowhere in the paper is mentioned how this solubility problem is solved even though in some experiments the concentration of 2h used is quite high. How do the authors control for the absence of precipitation?

Minor

- Authors should change the name of their drug. 2h is confusing: it is an abbreviation used for 2 hours
- The reference 65 is not correct (it is a tuberculosis paper)
- The Figures' legends could be clearer.

Reviewer #2 (Remarks to the Author):

General comment: this is a thorough work deciphering the mechanisms of action of a novel potential antifungal drug (2h), which inhibits a component of the Hsp110 family (Msi3) in *Candida albicans*. Overall, the scientific approach is rigorous and there is an extensive molecular work to characterize this molecule and its target.

The work could considerably gain in interest if the authors would have tested the activity of this compound against other *Candida* spp. that are notoriously more resistant to currently licensed antifungals, such as *Candida glabrata* or *Candida auris*. This is mainly against these pathogens that we need new drugs.

Another limitation is the absence of evidence that this compound would not be toxic for humans and has a real potential for future clinical application in terms of PK/PD profile. Indeed, there are many compounds with in vitro antifungal activity that do not reach the clinical stage. Molecular chaperones are known to be essential and highly conserved in eukaryotes, and therefore are difficult to target. The authors provide some arguments, such as the less conserved sequence of Hsp110 and the lack of growth inhibition in a human cell line (although we still see some inhibition at the effective concentration of the drug), but these are not robust evidences for lack of toxicity. Moreover, the in vivo efficacy of the drug relies on many parameters that are not evaluable in vitro (pharmacokinetic of the drug, rate of elimination, tissue penetration etc.) Some animal model (mice) would be essential to assess the real potential of this compound for future clinical application.

Introduction: too long, should be shortened.

Page 3 line 45 : « marginally effective ». I don't think we can consider current antifungals as « marginally » effective. Indeed, they have demonstrated their efficacy.

Page 3, lines 47-49: « Although several inhibitors were identified recently, the lack of clear targets and inhibiting mechanisms have prevented further development. » This sentence does not make really sense (inhibitors of what? Targets of the novels compounds in the antifungal pipeline, e.g. fosmanogepix, olorofim, ibrexafungerp, have clear targets and mechanisms of action).

Page 7, lines 235-236: why providing only MIC50 and not MIC90 or true MIC (complete growth inhibition) as the drug appears to be very potent and fungicidal ?

Page 7, lines 239-240: "growth was largely unaffected when using a human cell line."

Not sure... According to figure 4B, it seems that growth goes down to about 70%. Looking at the concentration at which 2h reaches its major effect (around 40 μ M), the relative HCT growth is still around 80%

Page 7, lines 257-260: "suggesting that 2h is fungicidal..." Time-kill curves should be done using usual criteria to define fungicidal vs fungistatic.

Figure 4, panel B: the legend is unclear. As growth of yeast cells and human cells are on a same graph, it should be clearly mentioned the type of cell + type of drug. For instance, red line should be: "C. albicans + 2h", yellow line should be: "HCT116 + 2h". Y axis: relative growth: compared to untreated conditions (?), specify. Note: there is still some growth inhibition (from 100% to around 70%) of human cells (HCT116) with 2h.

Antifungal susceptibility testing and synergy testing should also be performed with validated methods for clinical laboratories (e.g. CLSI or EUCAST protocols, and checkerboards for interactions).

Page 9. Line 325: polyenes (ampho B) do not target biosynthesis pathway of ergosterol (but binds to ergosterol leading to membrane destabilization and ion leakage).

Page 9. Lines 335-336: "A combination of 2h and available antifungals such as azoles may be the key to treating various candidiasis efficiently." Why a combination? Considering the potent in vitro effect, monotherapy might be sufficient.

Reviewer #3 (Remarks to the Author):

The Hsp110 molecular chaperone is conserved in and exclusive to eukaryotic cells where it plays at least two major roles: as an essential nucleotide exchange factor for the related Hsp70 chaperone family, and as a potent substrate stabilizing "holdase" that prevents protein aggregation but does not catalyze folding. As a critical part of the proteostasis network, there is considerable interest in identifying small molecules to both probe mechanistic aspects of Hsp110 function as well as to modulate overall proteostasis. Msi3 is the Hsp110 homolog in the pathogenic yeast *Candida albicans* and is essential for viability. Hu and coworkers have succeeded in identifying a potent small molecule ("2h") that appears to be fungistatic to *C. albicans* as well as the non-pathogenic yeast *Saccharomyces cerevisiae*, with the additional ability to inhibit hyphal formation in response to pathogenic cues. Mechanistic experiments convincingly, if somewhat surprisingly, demonstrate that 2h does not interfere with Hsp70 binding or Msi3 NEF activity. Instead, it appears to inhibit substrate binding (the holdase activity) in a manner lethal to the organism.

While I am very enthusiastic about the findings in this report, the results are both intriguing and puzzling. The molecule clearly kills *C. albicans* with limited toxicity to mammalian cells, a major accomplishment in its own right regardless of target. It is unclear at this time whether the hyphal inhibition phenotype is linked to the fungicidal mechanism since many viable hyphal-lacking mutants of *C. albicans* have been described. A more critical unanswered question is whether 2h is effective in reducing virulence of *C. albicans* in an animal model.

The major confounding result is the apparent targeting of the molecule to the SBD, or at least the substrate binding activity. Several reports over the last few years from the Bukau, Morano, and Andreasson laboratories (all cited within the manuscript) have found that the essential property of Hsp110 is the NEF activity, not the substrate holdase activity. Therefore, the current findings are inconsistent with the weight of past evidence and will require additional support. Is binding of 2h and the TRP2 peptide

competitive? Does 2h lock Msi3 into the "ATP" state instead of the extended conformation? Some additional mechanistic insight is needed to maximize the impact of this report.

Major comments:

- 1. A major thrust of this manuscript is the possible utility of 2h as a fungicidal agent to treat *C. albicans* infections. What's lacking in the current manuscript is the gold standard experiment of testing virulence in response to treatment with 2h in an animal model. This experiment is critical to establishing the potential impact of the story and should be achievable with a skilled collaborator in the field of candidiasis and virulence.**
- 2. The authors clearly show that 2h has no effect on bacterial protein refolding using the DnaK-DnaJ-GrpE system, but this is more of an orthologous rather than homologous comparison. A noted absence from this manuscript is testing of the effects of 2h on any of the three mammalian Hsp110 isoforms, Hsp105alpha, Apg-1 or Apg-2. These proteins have been successfully purified by several labs and the experiments are identical to those already done for Msi3. Results from these experiments are critical for pursuing 2h as a fungicidal agent; alternatively, they may reveal conservation of function between fungal and mammalian Hsp110 chaperones that can be further probed mechanistically using the small molecule.**
- 3. It is surprising that standard k_m/v_{Max} determinations were not performed to establish competitive vs. non-competitive modes of interaction of 2h with ATP in the Msi3 NBD or 2h with the TRP2 peptide in the SBD. It seems the authors may already have the data in hand or if not could perform quick concentration curve assays to allow these assessments to complement the existing findings.**
- 4. The Discussion does not go far enough, in my opinion, to postulate one or more testable mechanisms for 2h function. How is this molecule working?**
- 5. One known link between the observed hyphal morphogenesis phenotypes and prior Hsp110 work in yeast are the described roles for Sse1 and the Hsp90 system in the cell wall stress pathway. Namely, normally temperature sensitive mutations in several chaperones in *S. cerevisiae* can be suppressed by osmotic stabilization of the medium. Have the authors tried this with the *Candida* fungicidal assays? It may be that the ultimate cause of cell death is not lack of Hsp110 function, per se, but rather a specific defect in cell wall maintenance.**

Minor comments:

- 6. Many of the figures include tabular data derived from the plots that detract from the aesthetic of each figure. These data should be moved to a supplement or simply referenced in the text.**
- 7. The process schematics in Figs 3B and 4A are probably unnecessary as the experimental schemes are standard.**
- 8. Is there something special about the recently developed small molecule library chosen to screen? 1 hit out of only 23 molecules is a very high percentage for such a screen, suggesting a bias toward chaperone binding in the collection. More data should be provided in the text to clarify.**

Dear reviewers,

Thank you for reviewing our manuscript. We are encouraged by the positive reviews from you and are grateful for your constructive and insightful comments and suggestions that have significantly helped us improve our work. We have taken all the feedback seriously and made every effort to thoroughly address all your concerns. As you can see, we have carried out several major experiments and made substantial revisions to our manuscript, including but not limited to testing the suggested controls. With the end of my R01 grant in the middle of 2022, as well as a shortage of staff and difficulty in collaboration amid the COVID-19 pandemic, we kindly ask for your understanding of the additional time it has taken us to complete the suggested experiments. In response to your comments, we have made the following changes to our manuscript. We have highlighted the major changes made to the manuscript text in blue for easy identification. For the Figures and Supplementary Figures, the major changes are reflected in the legends (highlighted in blue font). We hope that we have addressed all the issues raised by you, significantly improved our manuscript, and made our work suitable for publication in *Nature Communications*.

We kindly request a prompt review of our manuscript. The primary reason for this request is that Dr. Cancan Sun, a first author and the major driving force behind this manuscript, is scheduled to leave our lab at the end of March due to funding shortages.

A. Reviewer #1:

Overall, the reviewer thinks our manuscript is “potentially interesting”. At the same time, the reviewer has raised important concerns about insufficient support for our conclusions (presented in blue font), which we believe have helped us to significantly improve our manuscript. We thank the reviewer for the insightful and constructive comments. Please find below our point-by-point response to the reviewer’s specific comments (presented in black font):

1. There is a big problem concerning the controls used to demonstrate the specificity of 2h. First, in all figures another chemical molecule should be included as a negative control. Second, a more relevant control than DNAK (a HSP70 protein from E coli) should be used. For instance, HSP70 from C albicans and other HSP110 proteins such as human HSP110 (it is very important to demonstrate that 2h does not affect human HSP110 if the authors want to conclude that 2h does not affect human cells).

Response: We thank the reviewer for bringing to our attention the two types of controls: 1) chemical molecules; and 2) fungal Hsp70 and human Hsp110. We have made every effort to perform both types of controls wherever possible.

1) Regarding the chemical controls, we have used three chemical molecules: Riociguat, Compound C, and fluconazole. 2H was designed and synthesized based on Riociguat and Compound C. Neither showed any antifungal effects (please see the updated Fig. S10). Their commercial availability in large amounts has made all the proposed control experiments possible. Please see

the revised manuscript: Fig. S3A for the ATP-FAM binding; Fig. S3B for the TRP2 peptide binding; Fig. S7B for the holdase activity; Fig. S9B-D for the refolding activity; and Fig. S10 for the growth test on SC5314.

2) Regarding fungal Hsp70 and human Hsp110 controls: (a) fungal Hsp70: we have cloned *C. albicans*' Ssa1, the major cytosolic Hsp70. However, the lack of apparent induction in *E.coli* has prevented purification. This is consistent with the poor expression of *S. cerevisiae*'s Ssa1 in *E.coli*. As described in our original manuscript, we have purified *S. cerevisiae*'s Ssa1 after expressing it in *Pichia pastoris* (the *Pichia pastoris* strain for expressing Ssa1 was kindly provided by Dr. Johannes Buchner). The Ssa1 proteins from *C. albicans* and *S. cerevisiae* share 84.9% sequence identity. Therefore, we have used *S. cerevisiae*'s Ssa1 as a fungal Hsp70 control in our manuscript. Please see Fig. 3C for the refolding activity and Fig. S1B and D for the binding activities of ATP and peptide substrate. (b) human Hsp110: We have purified Hsp105, a human Hsp110, and carried out the following controls: Fig. 3B for the holdase activity; Fig. S1A and C for the binding activities of ATP and peptide substrate; and Fig. S9A for the refolding activity.

2. To demonstrate the role of 2h in the cells' growth and pathogenesis, a transcriptomic analysis showing that genes involved in these two processes are affected by 2h is needed. Authors could also use a proteomic approach to study how 2h affects the interactome of Msi3. To say that the effect of 2h is through protein homeostasis (supported only by a single luciferase assay) is not enough. Many chaperones have an effect in proteins homeostasis and there is a lot of redundancy in this function essential for cell viability. In this case, this can be just a consequence of blocking Msi3, but not necessary the cause of the effect of 2h on cell growth and/or pathogenesis. Deciphering the effect of 2h on Msi3s' interactome would also help select a relevant Msi3 substrate (more relevant than the TRP2 peptide).

Response:

1) We have carried out a transcriptomic analysis as the reviewer suggested and included the results in the last subsection of the Results section in the revised manuscript. Accordingly, we have also updated Fig. 6 and added a new table (Table S1) and a supplementary figure (Fig. S14). Interestingly, we found that the transcription of Msi3 and related chaperones such as Ssa1 and Ydj1 is up-regulated, which is consistent with the inhibition of 2H on Msi3's chaperone activity.

2) In addition to the *in vivo* luciferase refolding assay, we have conducted two additional *in vivo* assays to analyze the effect of 2H on two essential processes associated with Msi3's function in maintaining protein homeostasis: solubilizing protein aggregates and overall protein folding. Both assays provide support for 2H's inhibition on protein homeostasis, which is consistent with the inhibitory effect of 2H on Msi3's *in vitro* chaperone activity. Please see the updated Fig. 5B and S13 in the revised manuscript. We have observed specific protein aggregates upon 2H treatment. Consistent with the reviewer's suggestion, these protein aggregates most likely represent endogenous substrates for Msi3, and inhibiting their folding by 2H could be the underlying cause of the observed cell death. Additionally, our biochemical analysis has identified Ulp1, a protein from *S. cerevisiae*, as a substrate for Msi3's holdase activity. 2H showed a similar inhibition on the holdase activity of Msi3 when Ulp1 was used as a substrate, suggesting that the inhibition of

2H on Msi3's holdase activity is not substrate specific. Please see the updated Fig. S6B in the revised manuscript.

3. Too few and too unclear results are shown to be able to conclude that the 2h molecule affects HSP110s' holdase activity without affecting its' NEF activity. For instance, in Figure 2C, although small, there is a difference between Msi3+ATP and Msi3+ATP+2h. And in Fig 3 B-D, I do not understand the choice of the DnaK chaperone machinery, a system that does not involve an HSP110 protein. Why should 2h have an effect? Why not use the HSP70 chaperone cycle (which uses HSP110) to study the effect of 2h? Does 2h bind to GrpE? Authors should use mutants of Msi3 (with no NEF or holdase activity) to prove what they are claiming.

Response: We have revised the manuscript to clarify this issue and thank the reviewer for pointing it out.

1) We agree that there is some small effect of 2H on the NEF activity (less than 20% reduction in k_{off}). To ensure the accuracy of our findings, we have revised the manuscript to reflect this small effect.

2) Regarding the control using the DnaK chaperone machinery in the original manuscript, we have removed it and updated our manuscript to include a human Hsp70-Hsp110 chaperone machinery. Please see the updated Fig. 3B and S9A.

3) Based on the reviewer's suggestion, we have tested a mutant Msi3, I164D, and showed that 2H inhibited its holdase activity. Please see the updated Fig. S6A in the revised manuscript. Our previous published data have demonstrated that the I164D mutation abolished the NEF activity while leaving the holdase largely intact. Moreover, we have identified Ulp1 as another substrate for Msi3's holdase activity, and 2H exhibited a similar inhibition effect (please see the updated Fig. S6B). We hope that these additional findings provide further support for the inhibition of 2H on the holdase activity of Msi3.

4. Fig 4: First of all, it seems that the authors are using a strain that is resistant to fluconazole. It has hardly any effect (the MCI50 for fluconazole seems to be over 50). No comparison between the two molecules is therefore possible. Authors could study both a resistant and a sensitive strain and the MCI80 should be presented.

Response:

1) We thank the reviewer for bringing up this important point. We have obtained SC5314, the most widely used *C. albicans* strain, and carried out growth and viability tests. We have included the results in Table 1 of the revised manuscript.

2) Based on the reviewer's suggestion, we have obtained and tested four fluconazole-resistant strains (JS14, JS15, FH5, and 12-99, kindly provided by Dr. Theodore White) in addition to the fluconazole-sensitive strain SC5314. We have included these results in Table 1 of the revised

manuscript. MIC₉₀ was presented based on our results and suggestions from both Reviewer #1 and #2.

5. Concerning the synergistic effect, the methodology used does not comply with the standards (Clinical Laboratory Standards Institute (CLSI) or European Committee on Antimicrobial Susceptibility Testing (EUCAST)). The interpretation of the results must take into account the fractional inhibitory concentration index (FICI).

Response: We thank the reviewer for pointing out this oversight. As suggested by both Reviewer #1 and #2, we have performed all the antifungal susceptibility and viability tests according to the guidelines set forth by the Clinical and Laboratory Standards Institute (CLSI). We have determined the fractional inhibitory concentration index (FICI) using the checkerboard (chequerboard) assay. Please see the revised Materials and Methods and Results sections.

6. Tolerance and resistance are not equivalent. Msi3 has been shown to be involved in tolerance (Nagao et al 2012). In this case, an impact of Msi3 on calcineurin signaling has been reported. Could this mechanism be involved in this synergy (if synergy there is)? An effect of 2h on protein (re) folding is proposed. Could 2h impact the target of fluconazole? The fluconazole efflux pump for example?

Response:

1) Using SC5314 and JS14 (a fluconazole-resistant strain), we have shown that there is no apparent synergy between 2H and fluconazole after determining the FICI using the checkerboard assay. Please see the subsection “**2H effectively reduces the growth of *Candida albicans* and is fungicidal.**” in the Results section of the revised manuscript. The previously observed synergy could be due to the strain that we used before (please see our response to the comment 4 above).

2) As the reviewer suggested, we have carried out a transcriptomic analysis on the *C. albicans* strain SC5314 and analyzed the impact of 2H on the target of fluconazole (*ERG11*). Fluconazole treatment has been shown to induce up-regulation of *ERG11* and related genes in ergosterol biosynthesis pathway such as *ERG1*, *ERG3*, and *ERG9*. None of these genes showed significant up-regulation upon 2H treatment, consistent with the different targets of 2H and fluconazole. Please see the updated Fig. S14A. In addition, *MDR1*, *CDR1*, and *CDR2* are multidrug efflux pump and transporters that confer resistance to numerous chemicals including fluconazole. Their transcription was up-regulated upon 2H treatment, consistent with the expected common response to compound treatments. Please see the updated Fig. S14B.

7. In Figure 4B, why use the colorectal cancer cell line HCT116? Authors cannot conclude just from the data shown in Fig 4B that 2h does not affect human cell growth. Other human cell lines, including non-cancerous cells, should be tested and shown in a different graph.

Response:

1) We used HCT116 because a previously published human Hsp110 inhibitor showed significant inhibition on this cell line. In contrast, 2H shows a limited effect on this cell line.

2) As the reviewer suggested, we have tested two non-cancerous human cell lines (HLF-1 and LX-2), and have included all the human cell lines in a new graph with SC5314 for comparison. Please see the updated Fig. 4A in the revised manuscript.

8. In fig 5, to study the fungicidal effect of 2h, MIC should be replaced by MFC. Furthermore, the effect of 2h combined to fluconazole should be included, using both a fluconazole resistant and a sensitive strain. And again, like in all the figures, a chemical compound that does not bind to Msi3 should be included as a negative control.

Response:

1) As the review suggested, we have determined the MFC values and included the data in Table 1 of the revised manuscript.

2) We have tested the fluconazole-sensitive strain SC5314 and the fluconazole-resistant strain JS14 and did not observe any obvious synergy between 2H and fluconazole. Please see the updated subsection “**2H effectively reduces the growth of *Candida albicans* and is fungicidal**” in the Results section.

3) As the reviewer suggested, we have tested two control compounds: Riociguat and Compound C. Please see the updated Fig. S10.

*9. Inhibition of pathogenicity is claimed based on the ability of *C albicans* to form filaments. Since this is an important conclusion, other tests should be performed, for example a biofilm formation inhibition test and/or an adhesion inhibition test.*

Response: As the reviewer suggested, we have performed a biofilm formation test. Please see the updated Fig. 4C-D. In addition, based on the insightful suggestions from Editor Dr. Sanchez, we have removed the claims regarding inhibition of pathogenicity and focused our manuscript on mechanistic understanding of Hsp110 as an important antifungal target using 2H. Consequently, we have revised the manuscript to better reflect the focus of our manuscript.

10. In Fig 6, rescue experiments could be included by addition of recombinant Msi3 (and human HSP110 as a negative control, if the authors prove in Fig 1 that 2h does not bind/affect human HSP110).

Response:

1) In the original Fig. 6, Msi3 was included in the *MSI3 sse1Δ* strain used. We have revised the manuscript to clarify this point.

2) We have taken the reviewer's suggestion and introduced Hsp105, a human Hsp110, into the *sse1Δ* strain. However, Hsp105 was unable to substitute for Sse1 despite significant expression. Please see the updated Fig. S8.

11. Finally, in the discussion it is mentioned that 2h has solubility issues. However, nowhere in the paper is mentioned how this solubility problem is solved even though in some experiments the concentration of 2h used is quite high. How do the authors control for the absence of precipitation?

Response: To address this concern, we have revised the last paragraph in the Discussion section on the solubility of 2H and included a new subsection titled “**Compound preparation and storage**” under the Materials and Methods section. As described, the concentration of 2H was kept below its solubility limit in all of our assays.

Minor

- Authors should change the name of their drug. 2h is confusing: it is an abbreviation used for 2 hours

Response: We have made the suggested change and thank the reviewer for this suggestion.

- The reference 65 is not correct (it is a tuberculosis paper)

Response: We have checked reference 65 and confirmed that it pertains to the chemical library that we used rather than a tuberculosis paper. We have thoroughly examined all the references and were unable to find any tuberculosis paper.

- The Figures' legends could be clearer.

Response: We have taken great care to revise the figure legends and provide as much clarity as possible.

B. Reviewer #2:

Overall, the reviewer thinks our manuscript is “a thorough work deciphering the mechanisms of action of a novel potential antifungal drug (2h), which inhibits a component of the Hsp110 family (Msi3) in *Candida albicans*”, further noting “the scientific approach is rigorous and there is an extensive molecular work to characterize this molecule and its target.” We thank the reviewer for these encouraging and positive comments on the strengths of our manuscript. At the same time, the reviewer has provided insightful and constructive comments to improve our manuscript (presented in blue font). Please find below our point-by-point response to the reviewer's specific comments (presented in black font):

1. The work could considerably gain in interest if the authors would have tested the activity of this compound against other Candida spp. that are notoriously more resistant to currently licensed antifungals, such as Candida glabrata or Candida auris. This is mainly against these pathogens that we need new drugs.

Response: We thank the reviewer for this constructive suggestion. We have tested *Candida glabrata* and confirmed a similar inhibition of 2H. Please see Table 1 in the revised manuscript.

2. Another limitation is the absence of evidence that this compound would not be toxic for humans and has a real potential for future clinical application in terms of PK/PD profile. Indeed, there are many compounds with in vitro antifungal activity that do not reach the clinical stage. Molecular chaperones are known to be essential and highly conserved in eukaryotes, and therefore are difficult to target. The authors provide some arguments, such as the less conserved sequence of Hsp110 and the lack of growth inhibition in a human cell line (although we still see some inhibition at the effective concentration of the drug), but these are not robust evidences for lack of toxicity. Moreover, the in vivo efficacy of the drug relies on many parameters that are not evaluable in vitro (pharmacokinetic of the drug, rate of elimination, tissue penetration etc.) Some animal model (mice) would be essential to assess the real potential of this compound for future clinical application.

Response: We thank the reviewer for this constructive suggestion. While we appreciate the importance of animal models for assessing the clinical potential of 2H, we believe that it is beyond the scope of our current manuscript, which focuses on providing mechanistic insights into the antifungal activity of 2H and its target, Hsp110s. Our aim is to establish proof-of-principle evidence for Hsp110s as an important target for novel antifungal drug design and to investigate the functions and molecular mechanisms of Hsp110s using 2H as a powerful tool. Overall, our results support the potential of 2H as a lead compound for developing novel antifungals by targeting Hsp110s; however, we recognize that further work is needed to improve its selectivity for pathogenic fungi and solubility. We believe that the suggested experiment using animal models would yield more meaningful results after we have developed improved compounds based on 2H in future studies.

To partially mitigate this concern on toxicity for humans, we have carried out additional tests on two human cell lines, as suggested by Reviewer #1. Please see the updated Fig. 4A.

3. Introduction: too long, should be shortened.

Response: We have shortened the Introduction as the reviewer suggested to improve the clarity and focus of our manuscript.

4. Page 3 line 45 : « marginally effective ». I don't think we can consider current antifungals as « marginally » effective. Indeed, they have demonstrated their efficacy.

Response: We have made the suggested change.

5. Page 3, lines 47-49: « Although several inhibitors were identified recently, the lack of clear targets and inhibiting mechanisms have prevented further development. » This sentence does not make really sense (inhibitors of what? Targets of the novels compounds in the antifungal pipeline, e.g. fosmanogepix, olorofim, ibrexafungerp, have clear targets and mechanisms of action).

Response: We thank the reviewer for pointing out our oversight. We have removed this sentence and revised the corresponding section based on the reviewer's comments.

6. Page 7, lines 235-236: why providing only MIC50 and not MIC90 or true MIC (complete growth inhibition) as the drug appears to be very potent and fungicidal?

Response: As both Reviewer #1 and #2 suggested, we have provided MIC₉₀ and MFC. Please see Table 1 in our revised manuscript.

7. Page 7, lines 239-240: “growth was largely unaffected when using a human cell line.” Not sure... According to figure 4B, it seems that growth goes down to about 70%. Looking at the concentration at which 2h reaches its major effect (around 40 uM), the relative HCT growth is still around 80%.

Response: Based on this comment, we have revised this section of our manuscript and changed the phrase “largely unaffected” to “limited impact”. Please see the subsection “**2H has limited impact on human cells but a fungicidal effect on *Candida glabrata***” in the revised manuscript.

8. Page 7, lines 257-260: “suggesting that 2h is fungicidal...” Time-kill curves should be done using usual criteria to define fungicidal vs fungistatic.

Response: We have determined time-kill curves as the reviewer suggested. Please see the updated Fig. 4B.

9. Figure 4, panel B: the legend is unclear. As growth of yeast cells and human cells are on a same graph, it should be clearly mentioned the type of cell + type of drug. For instance, red line should be: “*C. albicans* + 2h”, yellow line should be: “HCT116 + 2h”. Y axis: relative growth: compared to untreated conditions (?), specify. Note: there is still some growth inhibition (from 100% to around 70%) of human cells (HCT116) with 2h.

Response: We thank the reviewer for bringing this to our attention. We have updated this figure with the addition of two more human cell lines and revised the figure legends based on the reviewer's suggestion to include: 1) labeling each line as the type of cell + type of drug; and 2)

defining the relative growth of the Y-axis. Please see the updated Fig. 4A and legends. In response to the Note, please see our response to comment 7 above.

10. Antifungal susceptibility testing and synergy testing should also be performed with validated methods for clinical laboratories (e.g. CLSI or EUCAST protocols, and chequerboards for interactions).

Response: As both Reviewer #1 and #2 suggested, we have performed antifungal susceptibility testing according to CLSI protocols and synergy testing using the chequerboard (checkerboard) assay. Please see the updated Materials and Methods and Results sections.

11. Page 9. Line 325: polyenes (ampho B) do not target biosynthesis pathway of ergosterol (but binds to ergosterol leading to membrane destabilization and ion leakage).

Response: We thank the reviewer for pointing out this oversight. We have revised this part as the reviewer suggested.

12. Page 9. Lines 335-336: “A combination of 2h and available antifungals such as azoles may be the key to treating various candidiasis efficiently.” Why a combination? Considering the potent in vitro effect, monotherapy might be sufficient.

Response: Based on the reviewer’s comment and our current data, we have removed this sentence and revised the Discussion section accordingly.

C. Reviewer #3:

Overall, the reviewer is “very enthusiastic about the findings” in our manuscript and thinks that the identification of 2H as an antifungal in our manuscript is “a major accomplishment in its own right regardless of target”. In addition, the reviewer thinks that “Mechanistic experiments convincingly” support our inhibitor 2H specifically inhibits the holdase activity while leaving the NEF activity largely intact “in a manner lethal to” *C. albicans*. We thank the reviewer for these encouraging and positive comments on the strengths of our work. At the same time, the reviewer has provided insightful and constructive comments and suggestions to improve our manuscript (presented in blue font). Please find below our point-by-point response to the reviewer’s specific comments (presented in black font):

Major comments:

1. A major thrust of this manuscript is the possible utility of 2h as a fungicidal agent to treat C. albicans infections. What’s lacking in the current manuscript is the gold standard experiment of

testing virulence in response to treatment with 2h in an animal model. This experiment is critical to establishing the potential impact of the story and should be achievable with a skilled collaborator in the field of candidiasis and virulence.

Response: Based on the insightful suggestions from Editor Dr. Sanchez, we have removed the claims regarding inhibition of pathogenicity and focused our manuscript on the mechanistic understanding of Hsp110 as an important antifungal target using 2H. Accordingly, we have revised the manuscript, including the title, to better reflect this focus. As stated in the abstract, our study provides proof-of-principle evidence for supporting Hsp110s as an important target for designing novel and potent antifungal therapeutics, in addition to probing the functions and molecular mechanisms of Hsp110s using 2H as a powerful tool (as this reviewer pointed out in comment 2). We agree that testing virulence in response to treatment with 2H in an animal model would provide valuable information, but believe that this is beyond the scope of our current manuscript. Overall, our results support the potential of 2H as a lead compound for developing novel antifungals by targeting Hsp110s; however, we recognize that further work is needed to improve its selectivity for pathogenic fungi and solubility. We believe that the suggested experiment using animal models would yield more meaningful results after we have developed improved compounds based on 2H in future studies.

Regarding the final point made in this comment, the ongoing COVID-19 pandemic has complicated potential collaborations, but we will keep this in mind as new opportunities arise.

2. The authors clearly show that 2h has no effect on bacterial protein refolding using the DnaK-DnaJ-GrpE system, but this is more of an orthologous rather than homologous comparison. A noted absence from this manuscript is testing of the effects of 2h on any of the three mammalian Hsp110 isoforms, Hsp105alpha, Apg-1 or Apg-2. These proteins have been successfully purified by several labs and the experiments are identical to those already done for Msi3. Results from these experiments are critical for pursuing 2h as a fungicidal agent; alternatively, they may reveal conservation of function between fungal and mammalian Hsp110 chaperones that can be further probed mechanistically using the small molecule.

Response: Based on the reviewer's suggestion, we have purified human Hsp105alpha and tested the effect of 2H on its activities. Please see the updated Fig. 3B (for the holdase activity) and Fig. S9A (for the folding activity). We also thank the reviewer for the insightful suggestion on using 2H to further probe the function of Hsp110s mechanistically, which inspired us to carry out the *in vivo* protein aggregation assays to evaluate the effect of 2H on solubilizing protein aggregation and overall protein folding *in vivo* (please see the updated Fig. 5B and S13).

3. It is surprising that standard k_m/v_{Max} determinations were not performed to establish competitive vs. non-competitive modes of interaction of 2h with ATP in the Msi3 NBD or 2h with the TRP2 peptide in the SBD. It seems the authors may already have the data in hand or if not could perform quick concentration curve assays to allow these assessments to complement the existing findings.

Response: Following the reviewer's suggestion, we have performed a concentration curve assay on the holdase activity of Msi3 and determined its IC₅₀. As the holdase activity is not an enzymatic activity, we used IC₅₀ instead of k_m/v_{Max} to measure the inhibitory potency. Please see the updated Fig. 3B in the revised manuscript.

4. The Discussion does not go far enough, in my opinion, to postulate one or more testable mechanisms for 2h function. How is this molecule working?

Response: As per the reviewer's suggestion, we have revised the Discussion section to include possible mechanisms for 2H function. In our revised manuscript, we have analyzed protein aggregation *in vivo* to evaluate the impact of 2H on two essential processes in maintaining proteostasis: solubilizing protein aggregates and protein folding. Our data showed that 2H treatment resulted in the failure of solubilizing protein aggregates and enhanced protein aggregation with a pattern different from that caused by heat shock, suggesting a unique inhibition on protein folding *in vivo*. Please see the updated Fig. 5B and Fig. S13. These specific protein aggregates most likely represent the endogenous substrates for Msi3. Through inhibiting the folding of many cellular proteins, 2H treatment results in enhanced protein aggregation and the collapse of proteostasis, which eventually leads to cell death. Please see the revised Results and Discussion sections for more details.

5. One known link between the observed hyphal morphogenesis phenotypes and prior Hsp110 work in yeast are the described roles for Sse1 and the Hsp90 system in the cell wall stress pathway. Namely, normally temperature sensitive mutations in several chaperones in S. cerevisiae can be suppressed by osmotic stabilization of the medium. Have the authors tried this with the Candida fungicidal assays? It may be that the ultimate cause of cell death is not lack of Hsp110 function, per se, but rather a specific defect in cell wall maintenance.

Response: We thank the reviewer for this insightful suggestion. Based on the reviewer's suggestion, we have carried out growth and fungicidal assays on the wild-type *C. albicans* strain SC5314 using osmotic stabilization with 1 M sorbitol in the medium. As shown in the updated Table 1, little impact on MIC and MFC was observed, suggesting that suppression of specific defects in cell wall maintenance caused by Hsp110 inhibition is not sufficient to support cell growth. In addition, osmotic stabilization of the medium using 1 M sorbitol only partially suppresses the growth defect of the *SSE1* deletion strain in our study, suggesting that the suppression by osmotic stabilization may be strain specific for the previously published results. Please see the updated Fig. S15.

Minor comments:

6. Many of the figures include tabular data derived from the plots that detract from the aesthetic of each figure. These data should be moved to a supplement or simply referenced in the text.

Response: As the reviewer suggested, we have removed all the tabular data from the figures and relocated them to either the main text or the Supplementary Figures. Please see the revised manuscript for these modifications.

7. The process schematics in Figs 3B and 4A are probably unnecessary as the experimental schemes are standard.

Response: We have removed these schematics as the reviewer suggested.

8. Is there something special about the recently developed small molecule library chosen to screen? 1 hit out of only 23 molecules is a very high percentage for such a screen, suggesting a bias toward chaperone binding in the collection. More data should be provided in the text to clarify.

Response: The library used in our screen contains 23 structurally related small molecules that were designed and synthesized based on Riociguat and Compound C. Due to patent restrictions, we are unable to provide further information about the remaining molecules. We are very fortunate to identify 2H as a hit from screening this library, while our preliminary screens using a commercial library did not yield any significant hit. We have provided clarification in the text.

Thank you again for your consideration. We appreciate your effort and expertise in reviewing our manuscript and look forward to hearing from you soon.

Sincerely yours,

Qinglian Liu, PhD
Professor
Department of Physiology and Biophysics
Virginia Commonwealth University, School of Medicine
1220 East Broad Street, Room 2042
Richmond, VA 23298

Reviewer #1 (Remarks to the Author):

All in all, I am satisfied with the way the authors have answered our critics. However some concerns remain:

1. In Figure 1D there is a very important increase in the fluorescence intensity in the presence of Msi3+2H+ATP. In contrast, in Fig S4 there is hardly no difference in the relative fluorescence for Msi3+2H+ATP. This is confusing. If the explanation is because the Y-axis (vertical axis) in each figure is different (fluorescence intensity in one and relative fluorescence in the other), this should be homogenized so the two figures can be compared.
2. The synergistic study of 2H and fluconazole using a checkerboard mentioned in the text (page 8, lines 291-295) should be shown in a Figure.
3. There is a problem with Figure 4A since you cannot include in the same figure the effect of 2H on *Candida* growth and on human cells' survival. One separate Figure should be the growth of *Candida albicans* and it should include both fluconazole and 2H; the other Figure should include just the human cell lines.
4. In Figure 4B, why the threshold is 99% instead of 99.9% (the definition given as the minimum fungicidal concentration)?
5. Figure 6C should be better explained. They mention there is no significant changes in transcription related to other cellular processes but the Figure shows up and down regulations. An explanation is needed.

Minor:

- In the introduction, the two pathways that target available antifungal drugs should be mentioned to further point out the originality of this work.
- It is not clear what does mean "compounds designed based on Riociguat and Compound C"(page 4, lines 120-121).
- In the legend of Figure 2D, the concentration of 2H used should be indicated

Reviewer #2 (Remarks to the Author):

The comments and suggestions have been adequately addressed. Thank you.

Reviewer #3 (Remarks to the Author):

The authors have done an extraordinary amount of work addressing all three reviewers' major concerns, including obtaining and testing the 2H compound against the human Hsp110/Hsp70 system and repeating most of the fungal inhibition assays. The paper has been refocused to target more the biochemistry and function of *Candida* Hsp110 (Msi3), with reduced emphasis on fungal pathogenesis. As such, the work stands on its own better rather than trying unsuccessfully to cover too much ground.

Fig 5B needs some polishing - the MWM should be cropped out and the authors should note that while it is possible that the proteins remaining in the pellet are Msi3 targets, it is also possible that they are simply hyperabundant cellular proteins. Without mass spectrometric identification, difficult to say which.

Overall, this manuscript will be an important and impactful addition to the Hsp110/Hsp70 chaperone field.

Dear reviewers,

We would like to express our sincere appreciation for your swift review of our manuscript. We are delighted to hear that we have addressed your concerns and suggestions adequately, and we are grateful for your additional constructive and insightful comments and suggestions to further improve our work. We have taken the remaining concerns seriously and made every effort to address them comprehensively. In response to your comments, we have made the following changes to our manuscript. We hope that we have resolved all issues raised by you, significantly improved our manuscript, and made our work suitable for publication in **Nature Communications**.

A. Reviewer #1:

Overall, the reviewer is “satisfied with the way the authors have answered our critics.” At the same time, there are some remaining concerns (presented in blue font), which we believe have helped us further improve our manuscript. We thank the reviewer for the insightful and constructive comments. Please find below our point-by-point response to the reviewer’s specific comments (presented in black font):

1. In Figure 1D there is a very important increase in the fluorescence intensity in the presence of Msi3+2H+ATP. In contrast, in Fig S4 there is hardly no difference in the relative fluorescence for Msi3+2H+ATP. This is confusing. If the explanation is because the Y-axis (vertical axis) in each figure is different (fluorescence intensity in one and relative fluorescence in the other), this should be homogenized so the two figures can be compared.

Response: To stress the important increase in the fluorescence intensity and clarify the confusion, we have made two changes to Fig S4a:

- 1) We have added a new plot to show the relative fluorescence intensities by setting the peak value of 2H alone as 1. Please see the top panel of the revised Fig. S4a. This plot highlights the important increase of fluorescence intensity in the presence of Msi3+2H+ATP.
- 2) We have added an explanation to the figure legends of Fig S4a to clarify how relative fluorescence was calculated. The original plot of Fig S4a is changed to the bottom panel in the revised Fig S4a. For this plot, the goal is to directly compare the shifts of peak wavelengths among fluorescence spectra. Thus, the relative fluorescence of each spectrum was calculated by setting the peak value of the corresponding spectrum as 1.

2. The synergistic study of 2H and fluconazole using a checkerboard mentioned in the text (page 8, lines 291-295) should be shown in a Figure.

Response: As the reviewer suggested, we have included the checkerboard result in Figure 4c in the revised manuscript.

3. There is a problem with Figure 4A since you cannot include in the same figure the effect of 2H on Candida growth and on human cells’ survival. One separate Figure should be the growth of

Candida albicans and it should include both fluconazole and 2H; the other Figure should include just the human cell lines.

Response: We thank the reviewer for this constructive suggestion. We have made the suggested changes. Please see:

- 1) the updated Figure 4a for the growth of *Candida albicans* including both fluconazole and 2H;
- 2) the updated Figure 4f for just the human cell lines.

4. In Figure 4B, why the threshold is 99% instead of 99.9% (the definition given as the minimum fungicidal concentration)?

Response: We thank the reviewer for pointing out this oversight. We have updated Figure 4b and its legends with the threshold defined as 99.9%.

5. Figure 6C should be better explained. They mention there is no significant changes in transcription related to other cellular processes but the Figure shows up and down regulations. An explanation is needed.

Response: Based on the reviewer's suggestion, we have provided additional explanation on Figure 6c. Please see the revised manuscript.

Minor:

- In the introduction, the two pathways that target available antifungal drugs should be mentioned to further point out the originality of this work.

Response: We thank the reviewer for this constructive suggestion. We have made the suggested change.

- It is not clear what does mean "compounds designed based on Riociguat and Compound C"(page 4, lines 120-121).

Response: To clarify this issue, we have revised this sentence to "compounds designed based on the chemical structures of Riociguat and Compound C". Please see the revised manuscript.

- In the legend of Figure 2D, the concentration of 2H used should be indicated.

Response: We have made the suggested change to the legend of Figure 2d.

B. Reviewer #2:

The reviewer stated that “The comments and suggestions have been adequately addressed.” We thank the reviewer for promptly reviewing our manuscript and for the positive and encouraging comment.

C. Reviewer #3:

The reviewer thinks “The authors have done an extraordinary amount of work addressing all three reviewers' major concerns” and “Overall, this manuscript will be an important and impactful addition to the Hsp110/Hsp70 chaperone field.” We thank the reviewer for promptly reviewing our manuscript and for these encouraging and positive comments on our manuscript.

In addition, there is one remaining concern: *Fig 5B needs some polishing - the MWM should be cropped out and the authors should note that while it is possible that the proteins remaining in the pellet are Msi3 targets, it is also possible that they are simply hyperabundant cellular proteins. Without mass spectrometric identification, difficult to say which.*

Response: We thank the reviewer for these insightful and constructive suggestions to further improve our manuscript. We have made the suggested changes.

- 1) We have cropped out the MWM from the gel on the right to simplify the results.
- 2) We have added a sentence to the Result section: “It is possible that the enhanced aggregation of some of these proteins could mainly be due to their high cellular abundance.” Please see the revised manuscript.

Thank you again for your consideration. We appreciate your effort and expertise in reviewing our manuscript and look forward to hearing from you soon.

Sincerely yours,

Qinglian Liu, PhD
Professor
Department of Physiology and Biophysics
Virginia Commonwealth University, School of Medicine
1220 East Broad Street, Room 2042
Richmond, VA 23298